# Design, Synthesis, and Evaluation of Novel 3-Carboranyl-1,8-Naphthalimide Derivatives as Potential Anticancer Agents

**DOI:** 10.3390/ijms22052772

**Published:** 2021-03-09

**Authors:** Sebastian Rykowski, Dorota Gurda-Woźna, Marta Orlicka-Płocka, Agnieszka Fedoruk-Wyszomirska, Małgorzata Giel-Pietraszuk, Eliza Wyszko, Aleksandra Kowalczyk, Paweł Stączek, Andrzej Bak, Agnieszka Kiliszek, Wojciech Rypniewski, Agnieszka B. Olejniczak

**Affiliations:** 1Institute of Medical Biology, Polish Academy of Sciences, 106 Lodowa St., 93-232 Lodz, Poland; srykowski@cbm.pan.pl; 2Institute of Bioorganic Chemistry, Polish Academy of Sciences, 12/14 Z. Noskowskiego St., 61-704 Poznan, Poland; d_gurda@ibch.poznan.pl (D.G.-W.); mplocka@ibch.poznan.pl (M.O.-P.); agaw@ibch.poznan.pl (A.F.-W.); giel@ibch.poznan.pl (M.G.-P.); wyszkoe@ibch.poznan.pl (E.W.); kiliszek@ibch.poznan.pl (A.K.); wojtekr@ibch.poznan.pl (W.R.); 3Department of Molecular Microbiology, Faculty of Biology and Environmental Protection, University of Lodz, 12/16 Banacha St., 90-237 Lodz, Poland; aleksandra.strzelczyk@biol.uni.lodz.pl (A.K.); pawel.staczek@biol.uni.lodz.pl (P.S.); 4Department of Chemistry, University of Silesia, 9 Szkolna St., 40-007 Katowice, Poland; andrzej.bak@us.edu.pl

**Keywords:** naphthalimides, carborane, anticancer activity

## Abstract

We synthesized a series of novel 3-carboranyl-1,8-naphthalimide derivatives, mitonafide and pinafide analogs, using click chemistry, reductive amination and amidation reactions and investigated their in vitro effects on cytotoxicity, cell death, cell cycle, and the production of reactive oxygen species in a HepG2 cancer cell line. The analyses showed that modified naphthalic anhydrides and naphthalimides bearing *ortho*- or *meta*-carboranes exhibited diversified activity. Naphthalimides were more cytotoxic than naphthalic anhydrides, with the highest IC_50_ value determined for compound **9** (3.10 µM). These compounds were capable of inducing cell cycle arrest at G0/G1 or G2M phase and promoting apoptosis, autophagy or ferroptosis. The most promising conjugate **35** caused strong apoptosis and induced ROS production, which was proven by the increased level of 2′-deoxy-8-oxoguanosine in DNA. The tested conjugates were found to be weak topoisomerase II inhibitors and classical DNA intercalators. Compounds **33**, **34**, and **36** fluorescently stained lysosomes in HepG2 cells. Additionally, we performed a similarity-based assessment of the property profile of the conjugates using the principal component analysis. The creation of an inhibitory profile and descriptor-based plane allowed forming a structure–activity landscape. Finally, a ligand-based comparative molecular field analysis was carried out to specify the (un)favorable structural modifications (pharmacophoric pattern) that are potentially important for the quantitative structure–activity relationship modeling of the carborane–naphthalimide conjugates.

## 1. Introduction

1,8-Naphthalimides are a class of polycyclic imides consisting of π-deficient flat aromatic or heteroaromatic ring systems. These compounds have been used in biological and nonbiological applications and have mainly been tested as DNA intercalators and anticancer as well as antibacterial, antiviral, and analgesic agents [1]. They exert their antitumor activity through the inhibition of topoisomerase I/II enzymes, photoinduced DNA damage, induction of reactive oxygen species (ROS) production, and malfunctions of lysosome and mitochondria [2], receptor tyrosine kinases [3], and DNA and RNA synthesis [4]. Most of the naphthalimides exhibit fluorescence and are thus widely used in biological imaging [5], as fluorescent probes for the targeted sensing of ions [6], endogenous molecules [7], and cancer cells [8].

Among the most promising and well-described naphthalimides are mitonafide, pinafide, amonafide, and elinafide, which exhibit excellent antitumor activity. Mitonafide, amonafide, and elinafide have entered phase II clinical trials. Despite their effectiveness, the latter two are no longer in clinical trials due to unpredicted toxicity caused by one of their metabolites, *N*-acetylamonafide, and neuromuscular dose-limiting toxicity, respectively [1,9]. The discovery, development, and structure–activity relationships (SARs) of 1,8-naphthalimide derivatives as anticancer agents were summarized by Tandon et al. [10] and Tomczyk et al. [9] in their works. It has been revealed that the modification of the naphthalimide backbone at different positions (especially positions 3 and 4) had remarkable effects on the anticancer activity, DNA binding properties and spectroscopic properties of the compounds [11].

The biomedical application of carboranes (C_2_B_10_H_12_) (Figure 1) has been reviewed in several papers [12,13,14,15,16,17,18,19], particularly focusing on their properties that may be of importance in the design of biologically active compounds, which include the following: their ability of unique noncovalent interactions (ionic interaction, σ–hole interaction, dihydrogen bond formation); spherical or ellipsoidal geometry and rigid three-dimensional (3D) arrangement (which offer versatile platforms for 3D molecular construction); lipophilicity, amphiphilicity, or hydrophilicity (depending on the type of boron cluster used which allows modulating the pharmacokinetics and bioavailability); chemical stability as well as susceptibility to functionalization; bioorthogonality; stability in biological environments; abiotic origin; and high content of boron (important for boron neutron capture therapy, BNCT).

In our recent work, we described the methods used for the synthesis of naphthalimides modified with carborane or metallacarborane groups, as analogs of mitonafide (Figure 1). The cytotoxic properties of the obtained conjugates were investigated in the human cancer cell lines HepG2 and RPMI 2650, and the results showed that the type of boron cluster affected the various cytotoxic activities of the tested compounds. Moreover, flow cytometry analysis indicated that the naphthalimide boron cluster conjugate could effectively induce cell cycle arrest at G0/G1 phase and promote mainly apoptosis in the HepG2 cell line. However, the studied compounds were found to be rather weak classical DNA intercalators, compared to mitonafide, which indicates other types of interaction with DNA [20].

Based on the above results, we continued our research, and in this paper, we describe a method for synthesizing naphthalimide derivatives, containing 1,2-dicarba-*closo*- dodecaborane (*ortho*-carborane) or 1,7-dicarba-*closo*-dodecaborane (*meta*-carborane) at position 3 of the heteroaromatic skeleton, to investigate their anticancer activity and ability to induce cell death, cell cycle arrest, ROS production, and inhibition of human topoisomerase IIα. Studies on calf-thymus DNA (ct-DNA) were performed to evaluate the interaction of the synthesized derivatives with DNA. Additionally, we carried out a SAR-mediated similarity assessment of the property profile of the conjugates containing carborane group.

## 2. Results and Discussion

### 2.1. Chemistry

#### 2.1.1. Synthesis of Mitonafide and Pinafide Analogs Containing Carborane Clusters

The novel naphthalimide derivatives containing carborane clusters described in this study (compounds **8**–**11**, **17**–**20**, **27**–**30**, **33**–**36**, **39**–**42**, Figure 2, Figure 3, Figure 4 and Figure 5) were synthesized using the following methods: (1) copper(I)-catalyzed Huisgen–Meldal–Sharpless 1,3-dipolar cycloaddition of azides and alkynes (i.e., click chemistry) (Scheme 1 and Scheme 2); (2) reductive amination (Scheme 3); and (3) amidation reactions (Scheme 4).

Click chemistry is a very efficient and popular method to modify molecules [21,22]. Naphthalimide–carborane conjugates were synthesized via click chemistry using a standard procedure involving three steps. In the first step, suitable boron cluster donors (**4**, **5**) [23] were dissolved in a mixture of THF and water. In the second step, a naphthalic anhydride containing a terminal triple bond (**3**, **14**, Scheme 1 and Scheme 2), a catalytic amount of sodium ascorbate, and CuSO_4_⋅5H_2_O were added, and the reactions were performed at 35 °C for 2–4 h. After purification, modified anhydrides (**6**, **7**, **15**, **16**, Scheme 1 and Scheme 2) were obtained in yields ranging between 48% and 74%, with the lower yield for derivatives modified with *meta*-carborane. In the third step, the modified naphthalic anhydrides were transformed to mitonafide and pinafide analogs via a nucleophilic reaction with appropriate amine. The yield of the products **8**–**11** and **17**–**20** (Scheme 1 and Scheme 2) achieved after isolation and purification by column chromatography was in the range of 41–84%. The modified anhydrides **6**, **7**, **15**, and **16**, as well as modified naphthalimides **8**–**11** and **17**–**20**, were characterized by ^1^H-, ^13^C-, and ^11^B-NMR, FT-IR, MS, RP-HPLC (Appendix A (Electronic Appendix A, ESI)), and TLC.

It is worth adding that products **8**–**11** can be directly synthesized from mitonafide or pinafide derivatives bearing terminal triple bonds with boron cluster donors **4**, **5**, especially in the presence of tris[((1-benzyl)-1*H*-1,2,3-triazol-4yl)methyl]amine (TBTA) as a ligand to complex and “protect” copper(I) [24]. However, an advantage of the synthetic pathway described above is that one substrate (the anhydride **6** or **7**) gives rise to two target products–mitonafide and pinafide analogs.

Due to its synthetic merits as well as the ubiquitous presence of amines among biologically active compounds, reductive amination plays a dominant role in pharmaceutical and medicinal chemistry. It is characterized by operational ease and a wide toolbox of protocols and hence is considered one of the key approaches to C−N bond construction [25].

Treatment of 3-amino-1,8-naphthalic anhydride (**12**) [26], 3-amino-*N*-[2-(dimethyl- amino)ethyl]-1,8-naphthalimide (**21**), or 3-amino-*N*-[2-(N-pyrrolidinyl)ethyl]-1,8-naphthalimide (**22**) [26,27] with an appropriate aldehyde containing *ortho*-carborane (**23**) or *meta*-carborane (**24**) [28] in anhydrous THF or MeOH at 65–70 °C under an inert (Ar) atmosphere resulted in the corresponding Schiff bases **25**–**30**, but these could not be isolated due to their instability (Scheme 3). Compounds **31**–**36** were obtained by treating the modified Schiff bases **25**–**30** witH-NaBH_3_CN, followed by column chromatography.

Decaborane (B_10_H_14_), one of the principal boron hydride clusters, has been reported as a mild reducing agent. It is quite stable, easy to handle, and can be effortlessly removed after reaction [29]. The decaborane cluster is used in various types of reactions, such as reductive amination of aldehydes [30], reductive etherification [31], or even one-pot reduction/reductive amination witH-Nitro compounds [32]. It has been reported that the synthesis of compounds **32**, **35**, and **36** modified with an *ortho*-/*meta*-carborane cluster was carried out using decaborane as a reducing agent, which resulted in an expected product with a lower or similar yield compared to that obtained from the reaction using NaBH_3_CN. Therefore, this method was not included in the Materials and Methods section in this paper. The structure, purity, and homogeneity of compounds **31**–**36** were confirmed by ^1^H-, ^13^C-, and ^11^B-NMR, FT-IR, MS, RP-HPLC (Appendix A (ESI)), and TLC.

Formation of amide bonds is one of the most frequently encountered reactions in the synthesis of biologically active compounds [33]. We developed methods for synthesizing 3-aminonaphthalimide derivatives bearing carborane clusters **39**–**42** by the formation of amide bond between naphthalimide and carborane (Scheme 4).

Briefly, 3-amino-*N*-[2-(dimethylamino)ethyl]-1,8-naphthalimide (**21**) or 3-amino-*N*-[2-(*N*-pyrrolidinyl)ethyl]-1,8-naphthalimide (**22**) with 3-(1,2-dicarba- *closo*-dodecaboran-1-yl)propionic acid (**37**) [34] or 3-(1,7-dicarba-*closo*-dodeca- boran-1-yl)propionic acid (**38**) [35] was dissolved in anhydrous CH_2_Cl_2_, and then anhydrous benzotriazol-1-yl-oxytripyrrolidinophosphonium hexafluorophosphate (PyBOP) and (trimethylamine) TEA were added to the solution. After the reaction, the crude products were purified twice by column silica gel chromatography, followed by which conjugates **39**–**42** were obtained as a white solid in moderate or good yield as follows: 54% (**39**), 66% (**40**), 55% (**41**), and 49% (**42**). The structure, purity, and homogeneity of these compounds were confirmed by ^1^H-, ^13^C-, and ^11^B-NMR, FT-IR, MS, RP-HPLC (Appendix A (ESI)), and TLC. An alternative method for synthesizing compounds **39**–**42** was developed using *N*-succinimidyl active esters containing carborane clusters [36]. However, the products were obtained after 4–10 days at 37 °C with the maximum yield of 39%. Therefore, this method was not included in the Materials and Methods section in the paper.

#### 2.1.2. X-ray Structural Analysis

Each crystal structure contains one molecule of carborane–naphthalimide conjugate in the asymmetric unit. In **39**, this unit also contains one molecule of water, while in **41** the unit has two water molecules (Figure 2). The water molecules are well defined in the electron density maps and participate in the hydrogen-bonding networks. In the case of **41**, the two water molecules are involved in linking the NH group of one molecule with the N atom of the dimethylamine group from a neighboring molecule, and with the carbonyl oxygen atoms of another neighboring molecule. In the case of compound **39**, the single water molecule links the NH group of one molecule with the dimethylamine group of another molecule, while the carbonyl oxygen takes part in hydrogen bond formation with the CH group of the carborane cluster of yet another molecule (Figure 3). The donor–acceptor distance of the latter is 3.12 Å, which indicates that this is a relatively strong bond for a carbon atom acting as a donor. Our earlier studies have shown that carborane groups can participate in weak H-bonding interactions [37], and it was observed that the C–H group of free carboranes is acidic in nature [38]. In both crystal structures, the mitonafide moieties were found to exhibit ring stacking, while the carborane clusters formed their own zones in the crystal lattice.

In compound **41**, the aromatic interactions are more extensive and the carborane clusters also interact extensively with one another, whereas in **39** the division is less clear because the carborane clusters are involved in H-bonding.

### 2.2. Biological Investigation

#### 2.2.1. In Vitro Cytotoxic Activity

The obtained compounds were investigated for in vitro antitumor activity by examining their cytotoxic effects using the MTT tetrazolium dye assay [39,40] against the human cancer cell line HepG2 established from hepatocellular carcinoma. IC_50_ refers to the drug concentration (µM) required to inhibit cell growth by 50%. The IC_50_ values determined for the synthesized compounds are summarized in Table 1.

Generally, naphthalimides modified with carboranes (**8**–**11**, **17**–**20**, **33**–**36**, **39**–**42**) exhibited more cytotoxicity than naphthalic anhydrides containing carborane clusters (**6**, **7**, **15**, **16**, **31**, and **32**). A comparison analysis of the naphthalimides in the series in terms of their activity revealed that conjugates **8**–**11** synthesized via click reaction (triazole ring attached directly to the heteroaromatic system) were the most cytotoxic than the modified naphthalimides that were obtained using click reaction (**17**–**20**, triazole ring attached through an oxygen atom to the heteroaromatic system), reductive amination (**33**–**36**), or amidation (**39**–**42**). The pinafide analog containing *ortho*-carborane **9** was identified to be the most cytotoxic to the tested tumor cell line at a concentration as low as 3.10 µM. The pinafide analog **11** containing *meta*-carborane was slightly less cytotoxic with an IC_50_ value of 4.79 µM. The mitonafide analog modified with *ortho*-carborane **8** (IC_50_ = 4.33 µM) showed a moderately lower cytotoxic activity compared to compound **9**, but the highest activity among the mitonafide analogs modified with a carborane cluster (**10**, **17**, **19**, **35**, **39**, **41**). It is worth mentioning that compounds **8** and **9** modified with *ortho*-carborane and compound **11** modified with *meta*-carborane were more active than naphthalimides bearing *ortho*- or *meta*-carborane at the *N*-imide position (5.95 and 7.84 µM, respectively) [20].

Due to the presence of an additional oxygen atom in their structure, in comparison to conjugates **8**–**11**, modified naphthalimides **17**–**20** have showed significantly lower cytotoxic activity with an IC_50_ of 9.68–14.85 µM, with the highest value determined for mitonafide analog **17** bearing an *ortho*-carborane.

Naphthalimide–carborane conjugates **33**–**36** containing an alkane linker between the amine group at position 3 of the ring and the carborane cluster showed higher cytotoxic activity than conjugates **17**–**20**, but slightly lower activity compared to **8**–**11**. Mitonafide analog bearing *ortho*-carborane **33** was the most active among the synthesized compounds, with an IC_50_ value of 4.77 µM. Compounds **39**–**42** synthesized by amide bond formation displayed lower cytotoxic activity than **33**–**36**, because of the presence of a C=O group between the amine group and the carborane cluster. Their cytotoxic activity was around one to two times lower (IC_50_ = 6.77–12.59 µM) than compounds **33**–**36** (IC_50_ = 4.77–8.65 µM). Naphthalic anhydrides **15**, **16**, **31**, and **32** showed moderate cytotoxicity against HepG2 cells (IC_50_ = 40.81–67.78 µM), while **6** and **7** were not toxic (IC_50_ > 100 µM).

#### 2.2.2. Cell Cycle Analysis by Flow Cytometry

Cell cycle disorders such as phase arrest might be an important cause of inhibition of cancer cell growth and consequently the loss of cell viability [41]. Previous research showed that many drugs induced cell cycle arrest at the G2/M phase in cancer cells [42]. To reveal the mechanism behind the inhibitory effect of the synthesized compounds on cellular viability, we sought to examine the cell cycle regulation. For this purpose, HepG2 cells were exposed to compounds **6** (115 µM), **7** (104 µM), **8** (4 µM), **9** (3 µM), **10** (8 µM), **11** (5 µM), **15** (68 µM), **16** (61 µM), **17** (10 µM), **18** (15 µM), **19** (14 µM), **20** (12 µM), **31** (53 µM), **32** (42 µM), **33** (5 µM), **34** (8 µM), **35** (9 µM), **36** (6 µM), **39** (11 µM), **40** (13 µM), **41** (10 µM), and **42** (6 µM). The chosen concentration of each of these compounds corresponded to the whole IC_50_ value. Mitonafide and pinafide were used as reference compounds in this analysis. After exposure, HepG2 cells were examined by flow cytometry, and their DNA content was measured by PI staining. Based on the DNA content, it was found that compounds **6**, **9**, **10**, **16**–**20**, **33**, **34**, **40**, and **41** exerted different effects on cell cycle than mitonafide and pinafide (Figure 4 and Appendix A (ESI)) which induced cell cycle arrest at the S and G2M phases [20], respectively.

The tested compounds affected the cell cycle by increasing the percentage of the cells in G0/G1 by up to 73.5% (compound **18**) compared to the control (58.3%). Accumulation of cells in this phase delayed the progression of the cell cycle and the beginning of the S phase. Previous studies showed that naphthalimide derivatives that were modified with a carborane or metallacarborane cluster at the *N*-imide position also caused cell cycle arrest at the G0/G1 phase [20]. In this study, we found that conjugates **7**, **8**, **11**, **15**, **31**, **32**, **35**, **36**, **39**, and **42** arrested the cell cycle in G2M, similar to pinafide, of which compound **42** increased the number of cells in this phase by up to 28.1%.

#### 2.2.3. Oxidative Stress Measurement in HepG2 Cells by Flow Cytometry

To shed light on the mechanism responsible for the inhibitory effect of the compounds on cellular viability, we examined their ability to induce the production of ROS. ROS production has been proposed as one of the mechanisms by whicH-Naphthalimides and their derivatives induced cell cycle arrest and apoptosis on cancer cells, which was confirmed by the flow cytometry analysis of oxidative stress induction [2].

To confirm whether ROS were involved in the induction of cell cycle arrest by compound, the level of intracellular ROS was analyzed. HepG2 cells were cultured for 24 h with compounds **6** (57.5 µM), **7** (52 µM), **8** (2 µM), **9** (1.5 µM), **10** (4 µM), **11** (2.5 µM), **15** (34 µM), **16** (30.7 µM), **17** (4.9 µM), **18** (7.5 µM), **19** (7.2 µM), **20** (6 µM), **32** (20.4 µM), **39** (5.3 µM), **40** (6.3 µM), **41** (5.2 µM), and **42** (3 µM) (Figure 5 and Appendix A (ESI)). The selected concentration of each compound corresponded to half of its IC_50_ value. Mitonafide and pinafide were used as reference compounds in the analysis.

The intracellular level of ROS was analyzed by dual staining with H_2_DCFDA/PI. DCF green fluorescence was triggered in the presence of ROS proportional to the intensity of oxidative stress. The most potent ROS inducer, among conjugates **6**–**11**, **15**–**20**, **32**, **39**–**42**, was mitonafide. Conjugates **10**, **18**, **32**, and **39** were less effective. Furthermore, compounds **9**, **17**, **20**, and **40**–**42** were more promising compared to control.

#### 2.2.4. Analysis of 8-oxo-dG in HepG2 Cells

As compounds **31** and **33**–**36** exhibited autofluorescence (Appendix A (ESI)), the level of intracellular ROS was measured by determining the content of 2′-deoxy-8-oxoguanosine (8-oxo-dG) in the enzymatic DNA hydrolysates obtained from HepG2 cells treated with the tested compounds. 2′-Deoxyguanosine is known to be the most susceptible to oxidation among the four canonical nucleosides, and 8-oxo-dG is the major oxidation product in DNA [43]. Under normal conditions, a genome has one 8-oxo-dG molecule per 10^5^–10^6^ guanosines, corresponding to thousands of 8-oxo-dG molecules per single cell. In this study, the content of 8-oxo-dG was measured using HPLC-UV-ED. The number of 8-oxo-dG molecules per 10^6^ dG was calculated (Table 2). We found that all tested compounds significantly elevated the number of 8-oxo-dG molecules compared to untreated control. Moreover, treatment with compound **35** caused much higher oxidative disturbances in DNA than mitonafide [20] resulting in almost four times higher number of 8-oxo-dG molecules in the cells (107.34 ± 0.57 vs. 28.24 per 10^6^ dG, respectively).

#### 2.2.5. Apoptosis/Necrosis, Autophagy, and Ferroptosis Assays by Flow Cytometry

Apoptosis and necrosis are the two major processes leading to cell death. Of these, the former is characterized by specific morphological and biochemical features including chromatin condensation, cell shrinkage, activation of caspase, and the loss of mitochondrial membrane potential [44]. It was found, that naphthalimide derivatives could induce cell death through apoptosis in the tested HepG2 and Bel-7402 cells [45].

To investigate whether the tested compounds induced apoptosis in HepG2 cells, we incubated the cells witH-Naphthalimide–carborane cluster conjugates for 24 h and performed a flow cytometry analysis. The compounds that did not show autofluorescence were analyzed by dual staining using YO-PRO-1/PI, while those showing strong green autofluorescence (**31**, **33**–**36**) were analyzed using Alexa Fluor 647 annexin V conjugate staining. The concentration chosen for each compound corresponded to the whole IC_50_ values. Mitonafide and pinafide were tested as reference compounds in this analysis.

The results indicated that the conjugates **6**, **7**, **17**, **19**, **20**, and **39**–**42** mainly promoted the apoptosis mode of cell death (Figure 6A and Appendix A (ESI); Appendix A (ESI)). Compounds **17**, **19**, **39**, and **40** induced early apoptosis (expressed as a percentage of apoptotic cells), and only a few late apoptotic cells were seen. Incubation with compounds **20** and **41** induced mainly the early stages of apoptosis with a moderate level of late apoptosis (13.35% and 18.60%, respectively). Compounds **6**, **7**, and **42** more rapidly induced cell death, where numerous cells underwent late apoptosis (**42**) and necrosis (**6**, **7**).

Among the tested compounds, **31** and **33**–**36** were found to be apoptosis inducers. Compounds **33**, **35**, and **36** displayed strong apoptotic properties, but compound **35** was identified as the strongest proapoptotic promoter and induced apoptosis in nearly 70% of the treated cells after 24 h (Figure 6B and Appendix A (ESI)). This compound was also found to be a very potent inducer of oxidative stress (Table 2).

Interestingly, some of the modified naphthalimides (**8**–**11** and **18**) did not exhibit positive green fluorescence signal corresponding to apoptotic cells, although their applied concentration corresponded to the whole IC_50_ value. To investigate whether these conjugates induced another type of regulated cell death (autophagy) in the tested HepG2 cells, we performed flow cytometry analysis using Green Detection Reagent that selectively stained autophagic vacuoles. For a strong activation of positive signal of autophagy, HepG2 cells were incubated with rapamycin, a potent inhibitor of mTOR [46]. Autophagy plays an important role in cellular homeostasis and disease pathogenesis and is also one of the reasons for the inhibition of cell growth. Through this process, cytosolic components and organelles are delivered to lysosomes for degradation. Small chemical molecules that have the ability to modulate autophagy may have pharmacological value for the treatment of various diseases [47].

It has been reported that the analog NPC-16 (naphthalimide–polyamine conjugate) triggered both apoptosis and autophagy in HepG2 cells, and further autophagy facilitated cellular apoptosis. Furthermore, mTOR signal pathway was involved in NPC-16-mediated autophagy in HepG2 cells [45]. In this study, compounds **8**–**11** and **18** were detected as potent activators of autophagy (Figure 6C and Appendix A (ESI)). These conjugates increased fluorescence by 26–37% compared to the control, with the highest increase of fluorescence caused by conjugate **11**, while rapamycin increased fluorescence by 24%. Autophagy typically precedes or occurs along with apoptosis. We found that the conjugates tested as mediators in this study could promote or inhibit cell apoptosis in HepG2 cells at their higher concentration or with extended incubation time.

Unexpectedly, we observed that modified naphthalimides **15**, **16**, and **32** generated stronger than expected fluorescent signals during the apoptosis/necrosis analysis at a concentration corresponding to their whole IC_50_ value. Compound **32** was detected as a potent ROS inducer (Figure 5). Intense oxidative stress is a feature of ferroptosis, which is a relatively recently discovered type of programmed cell death and is usually accompanied by high iron accumulation and lipid peroxidation. Recent studies have shown that ferroptosis is closely related to the pathophysiological processes of many diseases and plays an important regulatory role in the development and progression of, for example, tumors, neurological disorders, acute kidney injury, and ischemia/reperfusion [48]. Therefore, we analyzed conjugates **15**, **16**, and **32** for their ability to induce ferroptosis. Cumene peroxide was used as a potent inducer of lipid peroxidation (positive control). The rate of lipid peroxidation was estimated using the reagent 581/591 C11 that localizes in the membranes of live cells. On the basis of red and green fluorescence data obtained by flow cytometry, we estimated the 590/510 ratio which is inversely proportional to the amount of peroxided lipids. The tested modified naphthalimides caused lipid peroxidation (Figure 6D and Appendix A (ESI)), although the highest peroxidation rate, indicated by the lowest 590/510 ratio, was found in the cells that were treated with conjugate **15**.

#### 2.2.6. Fluorescence Imaging of Lysosomes

Lysosomes, which are one of the vital organelles, participate in many physiological processes such as cell apoptosis, cell cycle progression, and supply of cellular energy. Recent research suggests that lysosomal dysfunction is a characteristic of autoimmune disorders and neurodegenerative diseases including lupus, rheumatoid arthritis, multiple sclerosis, and Alzheimer’s and Parkinson’s disease [49]. Due to the acidic feature of lysosomes, a series of lysosome-targeting probes of naphthalimide derivatives, in which the morpholine group is modified, have been designed and synthesized to enhance their targeting effects to the lysosomes through electronic interactions [50]. Lysosome-targeting anticancer agents based on naphthalimide derivatives are limited [51]. It was shown that *N*,*N*-bis(3-aminopropyl)methylamine-bridged bis-naphthalimide derivatives exhibited fluorescence imaging in lysosomes in HeLa cells [52]. Due to fluorescence selection of naphthalimide–carborane conjugates (Appendix A (ESI)), we studied the lysosome-targeting behavior and imaging capacity of compounds **33**, **34**, and **36** by performing co-localization experiments using a commercial lysosomal tracker (DND-99) as the co-localization agent. We observed that the fluorescence of compounds **33**, **34**, and **36** (green panels) in the co-stained cells overlapped well with that of DND-99 (red panels), as supported by their merged images (right panels) shown in Figure 7. This suggests that compounds **33**, **34**, and **36** can specifically target the lysosomes of living cells with good cell membrane permeability.

#### 2.2.7. Human Topoisomerase IIα Relaxation Assay

The substituted 1,8-naphthalimides can act as DNA intercalators stabilizing DNA–topoisomerase II complexes. Their interaction with DNA disrupts the cleavage–relegation equilibrium of Topo II, thus resulting in the formation of broken DNA strands [51,53].

Carborane cluster-modified naphthalic anhydrides (**6**, **7**, **15**, **16**, **31**, **32**) and naphthalimides (**8**–**11**, **17**–**20**, **33**–**36**, **39**–**42**) were tested in the screening assay for human topoisomerase IIα inhibitory activity, at a concentration of 100 µM (Appendix A (ESI)). The inhibitory activity, manifested as the presence of the supercoiled DNA fraction of pBR322 plasmid, was observed for compound **7**, and to some extent, for compound **6** (as the presence of a separate band of supercoiled plasmid as well as a discrete smear below the relaxed DNA fraction). Therefore, we subjected both compounds to further detailed analyses of inhibitory potential within the concentration range of 25–200 µM. As expected, compound **6** demonstrated relatively weak inhibitory effect on human topoisomerase IIα, with a maximum inhibitory activity of 19.8% at 100 µM (at 200 µM the inhibitory activity of the compound was slightly lower (14.5%) probably due to precipitation) (Figure 8A). Compound **7** acted in a concentration-dependent manner, and the best inhibitory activity was detected at 100 and 200 µM (19% and 44% inhibition, respectively) (Figure 8B). Noteworthy was the presence of a relatively significant amount of DNA in the gel wells, which increased in proportion with the concentration of compound **7**. One can assume that this fraction could have been generated either by additional inhibition of the enzyme’s decatenation activity or by the intercalation of the compound into a DNA helix leading to the formation of a covalent complex between the DNA and the enzyme, which cannot migrate in the agarose gel (no protein denaturant was added to the reaction) [54]. At the same time, a trapped topoisomerase could not relax supercoiled plasmids and hence the decrease in enzyme’s activity and the appearance of the supercoiled DNA band. The concentration-dependent inhibitory activity of compound **7** suggests that the used concentration might not be sufficient for the compound to intercalate into all plasmid molecules or that the binding of compound to the DNA might be reversible, resulting in an incomplete inhibition of the enzyme activity and the appearance of a relaxed DNA fraction. Considering that enzyme inhibition can be determined as a sum of the amount of supercoiled DNA and the DNA fraction trapped in the gel wells, we calculated 30% and 72% inhibition of topoisomerase activity for compound **7** at the concentration of 100 and 200 µM, respectively, and an IC_50_ value (concentration that inhibits the activity of an enzyme by 50%) of 134 µM.

### 2.3. Physicochemical Investigation with DNA

DNA melting is the process of separating the double-helical DNA into two single strands by disrupting the stable hydrogen bonding and base stacking interactions [55]. The melting temperature (*T*_m_) of DNA is defined as the point at which half of the DNA strands are in the double-helical state and the other half in a random-coil state [56]. DNA helix melting is performed by measuring the absorbance of DNA at 260 nm as a function of temperature. A large increase in *T*_m_ (3–8 °C) is observed only for the strong intercalation type of interaction, whereas groove-binding interaction of small molecules with DNA leads to insignificant amendment of *T*_m_. In our study, we conducted an experiment to monitor the changes in *T*_m_ for ct-DNA in the absence and presence of modified naphthalimides to understand the interaction between these compounds and ct-DNA (Table 3, Appendix A (ESI)).

Thermal melting experiments showed that the studied compounds **9**–**11**, **15**–**20**, **32**–**36**, **41**, and **42** caused negligible stabilization of ct-DNA, while conjugates **6**, **8**, **39**, and **40** caused better DNA stabilization. In comparison to mitonafide and pinafide (Table 3, Appendix A (ESI)), the conjugates rather excluded classical intercalation as a dominant binding mode, which indicates a different mode of interaction with DNA.

Some of the modified anhydrides, **7** and **31**, caused destabilization of ct-DNA. Drug-induced destabilization of DNA helix represents a novel antitumor mechanism of action and is associated with particular intercalation processes or postalkylation distortion of DNA. DNA-destabilizing compounds are relatively rare and constitute a minor proportion of DNA-interacting molecules (which primarily stabilize the double helix). Certain mono- or bis-intercalators and DNA alkylating agents exhibit such DNA-destabilizing effects. The formation of locally destabilized DNA portions could interfere with protein/DNA recognition and thus potentially affect several crucial cellular processes, such as DNA repair, replication, and transcription [57].

To better understand the interactions of DNA and modified naphthalimides, we conducted circular dichroism (CD) measurements. CD is a powerful and reliable technique to investigate the conformational changes in DNA morphology during interactions between a small molecule and DNA. The CD spectra of the B-form DNA duplex generally display a positive Cotton effect at 270 nm and a negative effect at approximately 250 nm, witH-Nearly equal magnitudes of longwave positive bands and shortwave negative bands [58,59]. The binding of a small, achiral molecule to a chiral DNA helix can result in induced CD signal from the molecule.

The CD spectrum of free ct-DNA showed a negative band at 248 nm due to polynucleotide helicity and a positive band at 276 nm due to base stacking, thus confirming the existence of ct-DNA in the right-band B-form [60]. As illustrated in Appendix A (ESI), treatment with mitonafide and pinafide caused a decrease in the negative peak and an increase in the positive peak. In contrast, conjugates bearing boron cluster **6**–**11**, **15**–**18**, **20**, **31**, **32**, and **34**–**36** did not cause any appreciable change in the CD spectra of ct-DNA (Appendix A (ESI)) with increase in concentration. In the case of compounds **19**, **33**, **39**, and **40**–**42**, the positive and negative bands were perturbed by the presence of these ligands (Appendix A (ESI)). This suggests that the compounds, especially **33** and **39**, interact strongly with DNA but slightly weaker than mitonafide and pinafide. Naphthalimides bearing boron clusters at the *N*-imide position also caused negligible stabilization of ct-DNA which was confirmed by the thermal melting experiment and CD spectra [20].

The interaction of naphthalimides containing carborane clusters was also studied by UV–vis absorption titration to better understand the mode of interaction and binding strength. Generally, bathochromic and hypochromic effects are observed in the absorption spectra if the small molecules intercalate with DNA [61]. The spectral changes observed in the electronic absorption of **6**–**11**, **15**–**20**, and **33**–**40** in the absence and presence of ct-DNA are illustrated in Appendix A (ESI). Progressive addition of ct-DNA at a concentration of 1.25–15 µM to a fixed amount of modified naphthalic anhydride or naphthalimide concentration (20 µM) caused a decrease in absorbance for almost all the tested compounds, with an exception of conjugate **6** for which an increase of absorbance was observed. The most significant decrease in absorbance was recorded for conjugates **8**, **10**, **17**–**20**, **39**, and **40** with the low concentrations of ct-DNA (from 0 to 2.50 µM). However, we did not observe a significant shift of absorption maxima and only a slight bathochromic shift of about 2 nm was observed for compounds **16** and **39**. This observation would rather indicate the groove binding of modified naphthalimide with ct-DNA, since insignificant (or small) shift in absorption spectral behavior (i.e., λ^abs^_max_) is generally accepted as the most probable consequence of groove binding [61]. On the other hand, the addition of ct-DNA to mitonafide caused a small bathochromic shift, which is also confirmed in the literature [62] (Appendix A (ESI)). A quantitative rationalization of the drug–DNA binding strength is important to evaluate the efficacy of a drug or a therapeutic agent. Therefore, to compare the DNA binding strength of the tested molecules, we calculated the binding constant K*_b_*, as described in the Materials and Methods section.

In comparison with the available literature reports on intercalative binding of strong intercalators such as ethidium bromide [62], we observed (Table 3) a lower binding constant by one (**6**–**10**, **16**–**20**, **33**–**39**) or two orders (**11**, **15**, **40**) of magnitude. However, the selected modified compounds (**6**–**10**, **16**–**20**, **33**–**39**) showed a similar K*_b_* value compared to mitonafide, and some of the tested compounds (**11**, **15**, **40**) revealed an analogous K*_b_* value to pinafide (Table 3). For conjugates **31**, **32**, **41**, and **42**, the K*_b_* value could not be determined due to the lack of noticeable changes in the UV spectra.

### 2.4. Similarity-Based Assessment of Property Profile

The concept of intermolecular guest–host recognition in the quantitative receptor-independent structure–activity modeling (RI-QSAR) stems loosely from the straightforward tenet of the substituent similarity [63]. In general, a congeneric series of molecules should exhibit similar pharmacological profile because the interchangeable groups characterized by similar size, shape, or electronic distribution are likely to induce similar effects on binding affinities (neighbor behavior). Despite some limitations, the search for distance-mediated similarity using a quantitative measure of the pairwise relatedness between two molecules, each with multidimensional (mD) pool of attributes, contributes favorably to the ligand-based SAR practice [64]. Comparative molecular field analysis (CoMFA) integrated with computational chemistry as an in silico procedure has long been established in the field of computer-assisted molecular design [65]. CoMFA specifies the molecular features in the form of steric and/or electrostatic ligand patterns for superimposed molecules using the spatial distribution of noncovalent areas evaluated over the lattice of points. Masking the explicit shape information by the regularity of the cubic grid lattice allows translating the structural data into spatially uniform maps of potential ligand–receptor interactions (pharmacophore) [66].

We conducted a similarity-guided property space assessment for the ensemble of carborane-containing conjugates using the principal component analysis (PCA). In addition, the enhancement of planar descriptor-driven projection with response data resulted in a structure–activity landscape with a subtle picture of (dis)allowed structural adjustment(s) potentially valid for molecular activities. Finally, CoMFA was employed in the quantitative SAR ligand-based study to indicate the steric and/or electrostatic features of the pharmacophore pattern.

We evaluated the similarity-driven property for the congeneric set of structurally related naphthalimide–carborane conjugates using PCA on the pool of 2361 descriptors retrieved from Dragon 6.0 program—constant or nearly constant values with a standard deviation (SD) of <10^−4^ were erased *a priori*. The mD data were organized into a matrix X_22×2361_ with rows and columns depicting molecules (objects) and descriptors (parameters), respectively. The standardized matrix was compressed effectively using PCA because the total variance described by the first three principal components (PCs) accounted for 82.87%, indicating that the parameters were highly intercorrelated. We scrutinized the 3D space defined by the first orthogonal components (PC1 vs. PC2 vs. PC3), which revealed that carborane-based derivatives are basically clustered into four subgroups as shown in Figure 9. As expected, the positional isomers (*ortho*-/*meta*-) in the borane cluster were positioned together.

Interestingly, the projection of the IC_50_ activity (expressed in the logarithmic scale) on the PC1 vs. PC2 plane clearly indicated the diagonal separation of the active (pIC_50_ > 4.5) and nonactive (pIC_50_ < 4.5) conjugates, as illustrated in Figure 10.

On the other hand, a similar tendency was not observed for the projection of objects on the two-dimensional (PC1 vs. PC2) space that was color-coded by Lipinski’s Rule of Five (Ro5) violations and molecular weights (MWs) accordingly. As a matter of fact, it was found that almost half of the analyzed carborane-containing conjugates did not strictly abide by Ro5 (Figure 11A) crossing the threshold value (MW ≤ 500) imposed on the MW descriptor (Figure 11B). Obviously, the violation of any two of the ADMET-friendly conditions reduces the probability of a compound to be orally bioavailable, whereas a good drug-like score does not make a molecule a drug (and vice versa) [67].

The gold standard of SAR-driven procedures is based on the similarity tenet, where the structural composition of chemicals influences their ADMET properties [68]. Despite the far-fetching over-simplification, the similarity concept conjugated with biological response is widely adopted in medicinal chemistry [69]. Conceptually, the pairwise descriptor-based structural relatedness between two objects can be quantitatively determined as a function of their common features, for instance using Tanimoto coefficient (T_c_) calculated for OpenBabel fingerprints. In this study, the distribution of this coefficient revealed a wide structural diversity of the analyzed molecules (ΔT_c_ ≈ 0.45) with the greatest frequencies recorded at 0.58 < T_c_ < 0.68, respectively, as depicted in Figure 12A. The lower values of T_c_ in the deltoidal matrix T_22×22_, as shown in Figure 12B, indicated the structural dissimilarities within the analyzed molecules, thus confirming our previous PCA findings (Figure 9 or Figure 10).

The conjugation of structural pairwise comparison with response affinity profile results in a graphical map for systematically investigating the SAR trends in the form of structure–activity landscape index (SALI) [70]. The specification of continuity areas and/or activity cliffs is related to the availability of structurally similar compounds characterized by noticeable variations in the biological response. In fact, even sparse sampling of the factual chemical space can roughly determine the SAR areas with sharply nonuniform regions (magic methyl phenomenon). A symmetrical SALI grayscaled heat map is presented in Figure 13A, which shows that the studied molecules are sorted correspondingly to their pIC_50_ values, with the legend representing the numerical SALI values.

Computationally, the structurally related molecules (e.g., positional stereoisomers, with T → 1) are characterized by SALI → infinity; therefore, such values were replaced by the highest (brightly color-coded) value of SALI. The right lower corner of the SALI plane (or the symmetrically positioned upper left one) was occupied by the most active molecules (**11**, **33**, **8**, **9**) accompanied by nonactive molecules (**6**, **7**, **15**, **16**), respectively. The bright SALI spots of the heat map in Figure 13A indicate that the specified molecules can potentially form activity cliff—Small structural variations are manifested via the demolition of activities.

Interestingly, the replacement of the anhydride-like fragment with the imide-based motif exerted a noticeable impact on molecular potency as revealed by our comparison between the most active and nonactive compounds (**11**, **33**, **8** versus **6**, **7**, **15**, **16**), respectively. Roughly speaking, it seems that the spatial arrangement of atoms/charges in the carborane cage (positional isomers) does not explain the variations observed in the biological response because nearly all active and nonactive molecules contain the same molecular borane-based scaffold attached to the hydrocarbon –(CH_2_)_3_– chain. In Figure 13B, the pairwise disparities of molecular activities are plotted against the structural (dis)similarities that are color-coded according to the SALI values. It appears that the sampling of structurally similar molecular pairs in the function of (un)favorable modifications (T ≥ 0.80 and ΔpIC_50_ ≥ 1) might be necessary to investigate the sparsely populated regions (rough areas) of the numerical SALI plane (the upper right corner in Figure 13B) and specify the SAR-related cliffs.

The modeling of biological/chemical effects of compounds and prediction of ADMET-based properties are challenging for contemporary in silico protocols. 3D-QSAR strategies, especially CoMFA, have greatly contributed to the specification of the spatial map of ligand–receptor interactions, namely pharmacophore mapping [71]. In fact, CoMFA allows constructing a spatially uniform 3D field around a series of superimposed molecules to investigate the molecular environment (steric/electronic features). Hence, the “indirect” ligand-based exploration of nonbinding fields (Lennard–Jones and Coulombic potentials) results in a 3D arrangement of the pharmacophoric properties of compounds sharing the common structural scaffold (chemotype). In this study, firstly, we subjected the superimposed training set of carborane-containing derivatives to CoMFA—the modeling of pIC_50_ data generated superior outcomes of the statistical metrics ($q_{cv}^{2}$ = 0.87) for the CH3+ atom probe. Obviously, the robustness of the CoMFA model and its predictive power are strongly related to the separation of training/test subsets [72]. On the other hand, there are no specific rules for the selection of training/test subgroups; therefore, we chose the Duplex algorithm to generate a 16:6 training/test (**8**, **15**, **20**, **31, 34**, **41**) population, respectively. Only a slight deterioration of the model performance was observed ($q_{cv}^{2}\;$= 0.83) with a low standard error of prediction (s = 0.244) for five optimal components. Interestingly, the results of CoMFA indicated that the steric field predominated the electrostatic one (fraction 0.82:0.18).

In reality, the direct translation of pharmacophore-based pattern into the corresponding pseudoreceptor model is not a trivial matter. In our study, the bundle of steric bulk was specified as privileged zones contributing (un)favorably to the ligand-based CoMFA model. Interestingly, the yellow 3D polyhedrals (Figure 14A) surrounding the carborane cage and the hydrocarbon chain attached with the triazole ring depicted the spatial areas marked as the unfavorable ones. In other words, the attachment of new substituents in the specified areas might have a detrimental impact on the activity of the carborane-containing compounds. On the other hand, the dominant green spheres in the close proximity of the pyrrolidine ring (Figure 14A) suggested that modifications in this area might favorably affect the carborane-based derivatives. It vaguely confirms the tendency observed for the majority of the pyrrolidine-based and tertiary amine-like carborane conjugates; the activity profile can be roughly ranked as pIC_50_ (pyrrolidine) > pIC_50_ (tertiary amine). It seems that a nearly constant arrangement of charges within the rigid carborane cluster for the entire set of investigated compounds resulted in a low contribution of electrostatic potentials to the CoMFA model. The performance of this CoMFA model is illustrated graphically in Figure 14B, which shows the plot of actual versus predicted pIC_50_ values for the training (blue dots) and test (red dots) sets, respectively. The selected test population covered uniformly the structural space of the analyzed compounds as shown in Figure 14B. Due to the general lack of correlation between high $q_{cv}^{2}$ and the predictive ability of mD-QSAR models, Golbraikh and Tropsha criterion ($q_{cv}^{2}$ > 0.5, $R^{2}>0.6,\;\left[\frac{R^{2}-R_{0}^{2}}{R^{2}}\right]<0.1,\;0.85\leq k\leq 1.15)\;$was applied for the CoMFA model, which resulted in satisfactory statistical metrics: $R^{2}=0.89,\;R_{0}^{2}=0.99,\left[\frac{R^{2}-R_{0}^{2}}{R^{2}}\right]=-0.1,\;k=0.99,\;SDEP=0.14,\;MAE=0.12,\;q_{test}^{2}=0.88$ [73].

## 3. Materials and Methods

### 3.1. Chemistry

Most of the chemicals were obtained from the Acros Organics (Geel, Belgium) and were used without further purification unless otherwise stated. Boron clusters were purchased from KATCHEM spol. s.r.o. (Řež/Prague, Czech Republic). All experiments that involved water-sensitive compounds were conducted under rigorously dry conditions and under an argon atmosphere. Flash column chromatography was performed on silica gel 60 (230–400 mesh, Sigma-Aldrich, Steinheim, Germany). *R*_f_ refer to analytical TLC performed using pre-coated silica gel 60 F254 plates purchased from Sigma-Aldric (Steinheim, Germany) and developed in the solvent system indicated. Compounds were visualized by use of UV light (254 nm) or a 0.5% acidic solution of PdCl_2_ in HCl/methanol by heating with a heat gun for boron-containing derivatives. The yields are not optimized.

^1^H-NMR, ^13^C-NMR, and ^11^B-NMR spectra were recorded on an Avance III 600 MHz spectrometer (Bruker, Billerica, MA, USA,) equipped with a direct ATM probe. The spectra for ^1^H, ^13^C, and ^11^B-Nuclei were recorded at 600.26 MHz, 150.94 MHz, and 192.59 MHz, respectively. Deuterated solvents were used as standards. The following abbreviations are used to denote the multiplicities: s = singlet, d = doublet, dd = doublet of doublets, ddd = doublet of doublets of doublets, t = triplet, dt = doublet of triplets, q = quartet, quin = quintet, bs = broad singlet, and m = multiplet. *J* values are given in Hz.

Mass spectra were recorded on a CombiFlash PurIon Model Eurus35 (Teledyne ISCO, Lincoln, NE, USA). The ionization was achieved by atmospheric-pressure chemical ionization (APCI) ionization in the positive ion mode (APCI+) and negative ion mode (APCI–). The entire flow was directed to the APCI ion source operating in the positive ion mode. Total ion chromatograms were recorded in the *m*/*z* range of 100 to 700. The vaporization and capillary temperature were set at 250–400 and 200–300 °C, respectively. Capillary voltage of 150 V, corona discharge of 10 µA. High-resolution mass spectra (HRMS) were obtained on an Agilent 6546 LC/Q-TOF with ESI ion source spectrometer (Agilent Technologies, Inc., Santa Clara, CA, USA). The data are presented for the most abundant mass in the boron distribution plot of the base peak (100%) and for the peak corresponding to the highest *m*/*z* value with its relative abundance (%).

The theoretical molecular mass peaks of the compounds were calculated using the “Show Analysis Window” option in the ChemDraw Ultra 12.0 program. The calculated *m*/*z* corresponds to the average mass of the compounds consisting of natural isotopes.

Infrared absorption spectra (IR) were recorded using a Nicolet 6700 Fourier-transform infrared spectrometer from Thermo Scientific (Runcorn, UK) equipped with an ETC EverGlo* source for the IR range, a Ge-on-KBr beam splitter, and a DLaTGS/KBr detector with a smart orbit sampling compartment and diamond window. The samples were placed directly on the diamond crystal, and pressure was added to make the surface of the sample conform to the surface of the diamond crystal.

UV measurements were performed using a GBC Cintra10 UV-VIS spectrometer (Dandenong, Australia). The samples used for the UV experiment were dissolved in 95% C_2_H_5_OH. The measurement was performed at ambient temperature.

RP-HPLC analysis was performed on a Hewlett-Packard 1050 system equipped with a UV detector, and Hypersil Gold C18 column (4.6 × 250 mm, 5 µm particle size, Thermo Scientific, Runcorn, UK). UV detection was conducted at λ = 340 nm. The flow rate was 1 mL min^–1^. All analyses were run at ambient temperature. The gradient elution was as follows: gradient A—10 min from 30% to 55% A, 10 min from 55% to 90% A, and 10 min from 90% to 30% A. Buffer A contained 0.1% HCOOH in CH_3_CN, and buffer B contained 0.1% HCOOH in H_2_O; gradient B—10 min from 0% to 25% A, 10 min from 25% to 60% A and 10 min from 60% to 0% A. Buffer A contained 0.1% HCOOH in CH_3_CN, and buffer B contained 0.1% HCOOH in H_2_O. Crystals of **39** and **41** were obtained by slow evaporation from MeOH. X-ray diffraction measurements on **39** were carried out under cryogenic conditions on beamline P13 equipped with PILATUS 6M detector, operated by EMBL Hamburg at the PETRA III storage ring (DESY, Hamburg, Germany). The data were processed using XDS [74], the structure was solved with SHELXT [75] and refined with SHELXL [76]. X-ray diffraction measurements on **41** were carried out under cryogenic conditions on SuperNova four-circle diffractometer (Oxford Diffraction, Abingdon, UK) equipped with a Cu anode and Atlas CCD detector. The data were processed with CRYSALISPRO software (Rigaku Oxford Diffraction) (Neu-Isenburg, Germany), the structure was solved with SHELXT and refined with SHELXL programs, as above, via the Olex^2^ interface [77]. The refinement of atomic positions was unrestrained except for hydrogen atoms which were maintained at riding positions. Appendix A (ESI) summarises the crystallographic data.

3-Iodonaphthalic anhydride (**1**) was synthesized as described in the literature [78]. Compound **1** was additionally purified by column chromatography on silica gel (230–400 mesh) using as an eluent CH_2_Cl_2_ to afford product as white solid. 1-(3-Azidopropyl)-1,2-dicarba-*closo*-dodecaborane (**4**) and 1-(3-azidopropyl)-1,7-dicarba- *closo*-dodecaborane (**5**) were synthesized as described in the literature [23]. 3-Aminonaphthalic anhydride (**12**) was synthesized as described in the literature [26]. 3-Hydroxynaphthalic anhydride (**13**) was obtained analogously to the synthesis of 4-hydroxy-naphthalic anhydride [79]. Compounds 3-amino-*N*-[2-(dimethylamino)ethyl]-1,8-naphthalimide (**21**) and 3-amino-*N*-[2-(*N*-pyrrolidinyl)ethyl]-1,8-naphthalimide (**22**) were obtained in two steps: (1) synthesis of 3-aminonaphthalic anhydride (**12**) [26], (2) reaction with the appropriate amine *N*,*N*-dimethylethylenediamine (for compound **21**) or *N*-(2-aminoethyl)pyrrolidine (for compound **22**) [27]. 2-(1,2-Dicarba-*closo*-dodecaboran-1-yl)ethanal (**23**) and 2-(1,7-dicarba-*closo*- dodecaboran-1-yl)ethanal (24) were synthesized as described in the literature [28]. 3-(1,2-Dicarba-*closo*-dodecaboran-1-yl)propionic acid (**37**) was synthesized as described in the literature [34]. *3*-(1,7-Dicarba-*closo*-dodecaboran-1-yl)propionic acid (**38**) was synthesized as described in the literature [35].

#### 3.1.1. Synthesis of 3-Ethynyl-1,8-Naphthalic Anhydride (**3**)

3-Iodo-1,8-naphthalic anhydride (**1**, 150 mg, 462.8 μmoL) was dissolved in anhydrous DMF (6 mL) and added to CuI (18 mg, 94.5 μmoL), and Pd(PPh_3_)_4_ (54 mg, 46.7 μmoL). Next, anhydrous TEA (130.5 μL, 93.6 μmoL) and trimethylsilylacetylene (263.5 μL, 1.85 mmoL) were added. Reaction mixture was stirred for 2 h at 65 °C under an inert (Ar) atmosphere. Subsequently, the solvents were evaporated to dryness under vacuum and 3-trimethylsilylethynyl-1,8-naphthalic anhydride (**2**) was purified by column chromatography on silica gel (230–400 mesh) using as an eluent CH_2_Cl_2_. Compound **2** (92 mg, 312.5 μmoL) was dissolved in mixture of TFA (5 mL), THF (800 μL) and H_2_O (800 μL). Reaction mixture was stirred for 8 h at RT and solvents were evaporated. The residue was dissolved in CH_2_Cl_2_ (5 mL) and solvent was evaporated. This was was repeated twice. Crude product was purified by column chromatography on silica gel (230–400 mesh) using as an eluent CH_2_Cl_2_ to afford product as white solid. Yield: 51 mg (50%). TLC (CHCl_3_): *R*_f_ = 0.40; ^1^H-NMR (acetone-d_6_, 600.26 MHz): *δ* (ppm) = 8.68 (d, 1H, *J* = 1.5 Hz, H_arom_), 8.62 (dd, 1H, *J* = 7.3, 1.1 Hz, H_arom_), 8.56 (d, 1H, *J* = 8.3 Hz, H_arom_), 8.52 (d, 1H, *J* = 1.5 Hz, H_arom_), 8.01 (dd, 1H, *J* = 8.3, 7.3 Hz, H_arom_), 4.04 (s, 1H, C-CH).

#### 3.1.2. Synthesis of 3-Prop-2-yn-1-yloxy-1,8-Naphthalic Anhydride (**14**)

3-Hydroxy-1,8-naphthalic anhydride (**13**, 131.3 mg, 613 μmoL), propargyl alcohol (38.9 μL, 674.5 μmoL) and PPh_3_ (193 mg, 735.8 μmoL) were suspended in anhydrous THF (1.3 mL). Suspension was cooled to 0 °C under an inert atmosphere (Ar), and solution of DIAD (144.5 μL, 735.8 μmoL) in anhydrous THF (5.3 mL) was added dropwise. Reaction mixture was stirred for 72 h at RT. Subsequently, water (6.5 mL) was added, and THF was evaporated. Crude product was extracted to CH_2_Cl_2_ (4 × 5 mL). The organic phase was separated, dried over MgSO_4_, filtered, and evaporated to dryness. The residue was purified by column chromatography on silica gel (230–400 mesh) using as an eluent CH_2_Cl_2_ to afford product as white solid. Yield: 108.4 mg (70%). TLC (CHCl_3_): *R*_f_ = 0.36; ^1^H-NMR (acetone-d_6_, 600.26 MHz): *δ* (ppm) = 8.45–8.44 (m, 2H, 2H_arom_), 8.21 (d, 1H, *J* = 2.5 Hz, H_arom_), 8.07 (d, 1H, *J* = 2.4 Hz, H_arom_), 7.90 (t, 1H, *J* = 7.8 Hz, H_arom_), 5.11 (d, 2H, *J* = 2.3 Hz, O-CH_2_), 3.22 (t, 1H, *J* = 2.3 Hz, C-CH).

#### 3.1.3. Synthesis of 1,8-Naphthalic Anhydride Derivatives **6**, **7**, and Naphthalimide Derivatives **8**–**11** Modified with Carborane Clusters via Click Reactions

1-(3-Azidopropyl)-carborane (*ortho*-carborane (**4**) or *meta*-carborane, (**5**)) (1 eqiuv.) was dissolved in mixture THF/H_2_O (2.5:1, *v/v*, 3 mL per 0.1 mmoL). 3-Ethynyl-1,8-naphthalic anhydride (**3**) (1 equiv.), CuSO_4_ 5H_2_O (0.05 equiv.) and sodium ascorbate (0.1 equiv.) were added. Reaction mixture was stirred for 2–4 h under argon at 35 °C. The reaction was quenched by evaporation of the solvents.

*3-{1-[3-(1,2-Dicarba-closo-dodecaborane-1-yl)propyl]-1H-1,2,3-triazol-4-yl}-1,8-naphthalic anhydride* (**6**): white solid, yield 12.2 mg (60%). TLC (MeOH/CH_2_Cl_2_, 1:49, *v*/*v*): *R*_f_ = 0.28; ^1^H-NMR (DMSO, 600.26 MHz): *δ* (ppm) = 8.98 (d, 1H, *J* = 1.2 Hz, H_arom_), 8.96 (s, 1H, CH_triazole_), 8.90 (d, 1H, *J* = 1.5 Hz, H_arom_), 8.58 (d, 1H, *J* = 8.1 Hz, H_arom_), 8.50 (d, 1H, *J* = 7.2 Hz, H_arom_), 7.93 (t, 1H, *J* = 7.8 Hz, H_arom_), 5.19 (br s, 1H, CH_carborane_), 4.47 (t, 2H, *J* = 7.0 Hz, CH_2_-triazole), 2.40–2.37 (m, 2H, CH_2_-carborane), 2.13–2.08 (m, 2H, CH_2_-C*H*_2_-CH_2_), 3.0–1.5 (m, 10H, B_10_H_10_); ^13^C-NMR (DMSO, 150.95 MHz): *δ* (ppm) = 160.49 (1C, C11), 160.46 (1C, C12), 144.67 (1C, C_triazole_), 135.36–119.08 (11C, 10C_arom_, CH_triazole_), 75.54 (1C, C_carborane_), 63.05 (1C, CH_carborane_), 48.63 (1C, CH_2_-triazole), 33.53 (1C, CH_2_-carborane), 29.26 (1C, CH_2_-*C*H_2_-CH_2_); ^11^B-NMR {^1^H BB} (DMSO, 192.59 MHz): *δ* (ppm) = −3.19 (s, 1B, B9), −6.19 (s, 1B, B12), −9.77 (s, 2B, B8, 10), −11.64–−13.11 (m, 6B, B3, 4, 5, 6, 7, 11); UV (99.8% EtOH): λ_max_ (nm) = 260, 266, 335, λ_min_ = 296, λ_sh_ = 320, 367; FT-IR: *ν*_max_ (cm^−1^) = 2964 (C-H_aliphat_), 2575 (B-H), 1770 (C=O), 1733 (C=O), 725 (B-B); RP-HPLC (gradient A): *t*_R_ = 18.23 min; APCI-MS: *m*/*z*: 450 [M + H]^+^, 492 [M + H + CH_3_CN]^+^, calcd for C_19_H_23_B_10_N_3_O_3_: 449.

*3-{1-[3-(1,7-Dicarba-closo-dodecaborane-1-yl)propyl]-1H-1,2,3-triazol-4-yl}-1,8-naphthalic anhydride* (**7**): white solid, yield 38.5 mg, (48%). TLC (MeOH/CH_2_Cl_2_, 1:49, *v*/*v*): *R*_f_ = 0.43; ^1^H-NMR (DMSO, 600.26 MHz): *δ* (ppm) = 8.93 (s, 2H, H_arom_ overlapped with CH_triazole_), 8.86 (d, 1H, *J* = 1.4 Hz, H_arom_), 8.54 (d, 1H, *J* = 8.1 Hz, H_arom_), 8.46 (d, 1H, *J* = 6.7 Hz, H_arom_), 7.90 (t, 1H, *J* = 7.7 Hz, H_arom_), 4.42 (t, 3H, *J* = 6.9 Hz, CH_2_-triazole), 4.05 (br s, 1H, CH_carborane_), 2.10–2.07 (m, 2H, CH_2_-carborane), 2.00–1.97 (m, 2H, CH_2_-C*H*_2_-CH_2_), 3.0–1.5 (m, 10H, B_10_H_10_); ^13^C-NMR (DMSO, 150.95 MHz): *δ* (ppm) = 160.41 (1C, C11), 160.40 (1C, C12), 144.57 (1C, C_triazole_), 135.32–118.98 (11C, 10C_arom_, CH_triazole_), 75.73 (1C, C_carborane_), 56.30 (1C, CH_carborane_), 48.75 (1C, CH_2_-triazole), 32.83 (1C, CH_2_-carborane), 29.92 (1C, CH_2_-*C*H_2_-CH_2_); ^11^B-NMR {^1^H BB} (DMSO, 192.59 MHz): *δ* (ppm) = −4.57 (s, 1B, B5), −11.09 (s, 5B, B4, 6, 9, 10, 12), −13.54 (s, 2B, B8, 11), −15.02 (s, 2B, B2, 3); UV (99.8% EtOH): λ_max_ (nm) = 261, 266, 334, λ_min_ = 296, λ_sh_ = 234, 323, 365; FT-IR: *ν*_max_ (cm^−1^) = 2956 (C-H_aliphat_), 2594 (B-H), 1769 (C=O), 1741 (C=O), 724 (B-B); RP-HPLC (gradient A): *t*_R_ = 16.80 min; APCI-MS: *m*/*z*: 450 [M + H]^+^, 492 [M + H + CH_3_CN]^+^, calcd for C_19_H_23_B_10_N_3_O_3_: 449.

*3-{1-[3-(1,2-Dicarba-closo-dodecaborane-1-yl)propyl]-1H-1,2,3-triazol-4-yl}-1,8-naphthalic anhydride* (**6**) or *3-{1-[3-(1,7-dicarba-closo-dodecaborane-1-yl)propyl]-1H-1,2,3-triazol-4-yl}-1,8-naphthalic anhydride* (**7**) (1 equiv.) was dissolved in absolute EtOH (10 mL per 0.45 mmoL) and *N,N*-dimethylethylenediamine (1.1 equiv.) (for compound **8**) or *N*-(2-aminoethyl)pyrrolidine (1.1 equiv.) (for compound **9**) was added. The reaction mixture was stirred for 1 h at 35 °C and then for 1 h at 45 °C under an inert (Ar) atmosphere. Subsequently, the solvent was evaporated to dryness under vacuum and crude product was purified by column chromatography on silica gel (230–400 mesh) with a gradient of MeOH (0–10%) in CH_2_Cl_2_. Additionally (only for compound **11**), purified product was dissolved in CHCl_3_ (1 mL) and poured into hexane (5 mL). A precipitate was isolated by centrifugation.

*N-{2-(Dimethylamino)ethyl]-3-[1-(1,2-dicarba-closo-dodecaborane-1-yl)propyl]-1H-1,2,3-triazol-4-yl}-1,8-naphthalimide* (**8**): yellow solid, yield 9.5 mg (41%). TLC (MeOH/CH_2_Cl_2_, 1:9, *v*/*v*): *R*_f_ = 0.29; ^1^H-NMR (acetone-d_6_, 600.26 MHz): *δ* (ppm) = 8.86 (d, 1H, *J* = 1.6 Hz, H_arom_), 8.76 (d, 1H, *J* = 1.5 Hz, H_arom_), 8.69 (s, 1H, CH_triazole_), 8.45 (dd, 1H, *J* = 7.2, 1.1 Hz, H_arom_), 8.37 (d, 1H, 8.2 Hz, H_arom_), 7.83 (dd, 1H, *J* = 8.1, 7.3 Hz, H_arom_), 4.73 (br s, 1H, CH_carborane_), 4.61 (t, 2H, *J* = 6.8 Hz, CH_2_-triazole), 4.28 (t, 2H, *J* = 6.9 Hz, CH_2_-N(CO)_2_), 2.67 (t, 2H, *J* = 6.9 Hz, C*H*_2_-N(CH_3_)_2_), 2.58–2.55 (m, 2H, CH_2_-carborane), 2.33–2.30 (m, 8H, N(CH_3_)_2_ overlapped with CH_2_-C*H*_2_-CH_2_), 3.0–1.5 (m, 10H, B_10_H_10_); ^13^C-NMR (acetone-d_6_, 150.95 MHz): *δ* (ppm) = 164.60 (1C, C11), 164.54 (1C, C12), 146.82 (1C, C_triazole_), 133.26–123.84 (11C, 10C_arom_, CH_triazole_), 76.54 (1C, C_carborane_), 63.82 (1C, CH_carborane_), 57.95 (1C, *C*H_2_-N(CH_3_)_2_), 50.13 (1C, CH_2_-triazole), 46.14 (2C, 2 × CH_3_), 38.92 (1C, CH_2_-N(CO)_2_), 35.55 (1C, CH_2_-carborane), 30.89 (1C, CH_2_-*C*H_2_-CH_2_); ^11^B-NMR {^1^H BB} (acetone-d_6_, 192.59 MHz): *δ* (ppm) = −2.80 (s, 1B, B9), −5.95 (s, 1B, B12), −9.59 (s, 2B, B8, 10), −11.53–−11.80 (m, 4B, B3, 4, 5, 6), −12.98 (s, 2B, B7, 11); UV (99.8% EtOH): λ_max_ (nm) = 257, 338, λ_min_ = 295, λ_sh_ = 231, 321, 372; FT-IR: *ν*_max_ (cm^−1^) = 2942 (C-H_aliphat_), 2574 (B-H), 1698 (C=O),1656 (C=O), 722 (B-B); RP-HPLC (gradient B): *t*_R_ = 22.28 min; APCI-MS: *m*/*z*: 520 [M+H]^+^, calcd for C_23_H_33_B_10_N_5_O_2_ = 519; HRMS (ESI+) 520.3867 [M + H]^+^, calcd for C_23_H_33_B_10_N_5_O_2_ = 520.3711 [M + H]^+^.

*N-[2-(N-Pyrrolidinyl)ethyl]-3-[1-(1,2-dicarba-closo-dodecaborane-1-yl)propyl]-1H-1,2,3-triazol-4-yl}-1,8-naphthalimide* (**9**): yellow solid, yield 14.3 mg (59%). TLC (MeOH/CH_2_Cl_2_, 1:9, *v*/*v*): *R*_f_ = 0.27; ^1^H-NMR (acetone-d_6_, 600.26 MHz): *δ* (ppm) = 8.84 (d, 1H, *J* = 1.5 Hz, H_arom_), 8.74 (s, 1H, H_arom_), 8.68 (s, 1H, CH_triazole_), 8.44 (d, 1H, *J* = 7.2, Hz, H_arom_), 8.34 (d, 1H, *J* = 8.2 Hz, H_arom_), 7.81 (t, 1H, *J* = 7.8 Hz, H_arom_), 4.73 (br s, 1H, CH_carborane_), 4.60 (dd, 2H, *J* = 8.3, 5.4 Hz, CH_2_-triazole), 4.33 (t, 2H, *J* = 6.8 Hz, CH_2_-N(CO)_2_), 2.95 (t, 2H, *J* = 6.8 Hz, CH_2_-pyrrolidine), 2.79 (br s, 4H, N-C*H*_2pyrrolidine_-CH_2_), 2.58–2.56 (m, 2H, CH_2_-carborane), 2.34–2.28 (m, 2H CH_2_-C*H*_2_-CH_2_), 1.80 (br s, 4H, CH_2_-C*H*_2pyrrolidine_-CH_2_), 3.0–1.5 (m, 10H, B_10_H_10_); ^13^C-NMR (acetone-d_6_, 150.95 MHz): *δ* (ppm) = 164.46 (1C, C11), 164.40 (1C, C12), 146.60 (1C, C_triazole_), 134.95–122.81 (10C, 10C_arom_, CH_triazole_), 76.36 (10C, C_carborane_), 63.64 (1C, CH_carborane_), 54.93 (1C, N-*C*H_2pyrrolidine_-CH_2_), 54.23 (1C, CH_2_-pyrrolidine), 49.94 (1C, CH_2_-triazole), 39.41 (1C, *C*H_2_-N(CO)_2_), 35.36 (1C, CH_2_-carborane), 30.68 (1C, CH_2_-*C*H_2_-CH_2_), 24.29 (1C, CH_2_-*C*H_2pyrrolidine_-CH_2_); ^11^B-NMR {^1^H BB} (acetone-d_6_, 192.59 MHz): *δ* (ppm) = −2.80 (s, 1B, B9), −5.95 (s, 1B, B12), −9.60 (s, 2B, B8, 10), −11.54 (s, 4B, B3, 4, 5, 6), −13.03 (s, 2B, B7, 11); UV (99.8% EtOH): λ_max_ = 257, 338 nm, λ_min_ = 296 nm, λ_sh_ = 238, 323, 372 nm; FT-IR: *ν*_max_ (cm^−1^) = 2960 (C-H_aliphat_), 2579 (B-H),1698 (C=O), 1659 (C=O), 723 (B-B), RP-HPLC (gradient B): *t*_R_ = 21.84 min; APCI-MS: *m*/*z*: 547 [M + H]^+^, calcd for C_25_H_35_B_10_N_5_O_2_ = 546.

*N-{2-(Dimethylamino)ethyl]-3-[1-(1,7-dicarba-closo-dodecaborane-1-yl)propyl]-1H-1,2,3-triazol-4-yl}-1,8-naphthalimide* (**10**): yellow solid. Yield: 19.3 mg (84%). TLC (MeOH/CH_2_Cl_2_, 1:9, *v*/*v*): *R*_f_ = 0.33; ^1^H-NMR (DMSO-d_6_, 600.26 MHz): *δ* (ppm) = 8.89 (s, 1H, CH_triazole_), 8.85–8.83 (m, 2H, 2H_arom_), 8.45–8.41 (m, 2H, 2H_arom_), 7.84 (t, 1H, *J* = 7.7 Hz, H_arom_), 4.42 (t, 2H, *J* = 6.7 Hz, CH_2_-N(CO)_2_), 4.16 (t, 2H, *J* = 6.8 Hz, CH_2_-triazole), 4.05 (br s, 1H, CH_carborane_), 2.53 (t, 2H, *J* = 6.8 Hz, C*H*_2_-N(CH_3_)_2_), 2.22 (s, 6H, 2CH_3_), 2.09–2.07 (m, 2H, CH_2_-carborane), 1.99–1.97 (m, 2H, CH_2_-C*H*_2_-CH_2_), 3.0–1.5 (m, 10H, B_10_H_10_); ^13^C-NMR (DMSO-d_6_, 150.95 MHz): *δ* (ppm) = 163.17 (1C, C11), 163.11 (1C, C12), 144.98 (1C, C_triazole_), 134.25–121.97 (11C, 10C_arom,_ CH_triazole_), 75.74 (1C, C_carborane_), 56.42 (1C, *C*H_2_-N(CH_3_)_2_), 56.30 (1C, CH_carborane_), 48.72 (1C, CH_2_-triazole), 45.30 (2C, 2 × CH_3_), 37.58 (1C, CH_2_-N(CO)_2_), 32.84 (1C, CH_2_-carborane), 29.95 (1C, CH_2_-*C*H_2_-CH_2_); ^11^B-NMR {^1^H BB} (DMSO-d_6_, 192.59 MHz): *δ* (ppm) = −4.62 (s, 1B, B5), −11.11 (s, 5B, B4, 6, 9, 10, 12), −13.58 (s, 2B, B8, 11), −15.05 (s, 2B, B2, 3); UV (99.8% EtOH): λ_max_ = 257, 339 nm, λ_min_ = 2978 nm, λ_sh_ = 234, 373 nm; FT-IR: *ν*_max_ (cm^−1^) = 2953 (C-H_aliphat_), 2577 (B-H), 1698 (C=O), 1655 (C=O), 722 (B-B); RP-HPLC (gradient B): *t*_R_ = 22.20 min; APCI-MS: *m*/*z*: 520 [M + H]^+^, calcd for C_23_H_33_B_10_N_5_O_2_ = 519.

*N-[2-(N-Pyrrolidinyl)ethyl]-3-[1-(1,7-dicarba-closo-dodecaborane-1-yl)propyl]-1H-1,2,3-triazol-4-yl}-1,8-naphthalimide* (**11**): white solid. Yield: 15 mg (55%). TLC (MeOH/CH_2_Cl_2_, 1:9, *v*/*v*): *R*_f_ = 0.34; ^1^H-NMR (CDCl_3_, 600.26 MHz): *δ* (ppm) = 8.83 (d, 1H, *J* = 1.5 Hz, H_arom_), 8.79 (d, 1H, *J* = 1,7 Hz, H_arom_), 8.54 (dd, 1H, *J* = 7.2, 0.8 Hz, H_arom_), 8.22 (d, 1H, *J* = 7.9 Hz, H_arom_), 8.00 (s, 1H, CH_triazole_), 7.74 (t, 1H, *J* = 7.7 Hz, H_arom_), 4.43 (t, 2H, *J* = 7.2 Hz, CH_2_-N(CO)_2_), 4.40 (t, 2H, *J* = 6.2 Hz, CH_2_-triazole), 2.99 (br s, 2H, CH_2_-pyrrolidine), 2.93 (br s, 1H, CH_carborane_), 2.88 (br s, 4H, N-C*H*_2pyrrolidine_-CH_2_), 2.10–2.07 (m, 4H, CH_2_-C*H*_2_-CH_2_ overlapped with CH_2_-carborane), 1.88 (br s, 4H, CH_2_-C*H*_2pyrrolidine_-CH_2_), 3.0–1.5 (m, 10H, B_10_H_10_); ^13^C-NMR (CDCl_3_, 150.95 MHz0): *δ* (ppm) = 164.18 (1C, C11), 164.15 (1C C12), 146.58 (1C, C_triazole_), 134.45–120.71 (11C, 10C_arom_, CH_triazole_), 74.72 (1C, C_carborane_), 55.21 (1C, CH_carborane_), 54.47 (1C, N-*C*H_2pyrrolidine_-CH_2_), 53.61 (1C, CH_2_-pyrrolidine), 49.80 (1C, CH_2_-triazole), 38.70 (1C, *C*H_2_-N(CO)_2_), 33.83 (1C, CH_2_-carborane), 30.53 (1C, CH_2_-*C*H_2_-CH_2_), 23.79 (1C, CH_2_-*C*H_2pyrrolidine_-CH_2_); ^11^B-NMR {^1^H BB} (CDCl_3_, 192.59 MHz): *δ* (ppm) = −4.18 (s, 1B, B5), −9.62 (s, 1B, B12), −10.59–−10.98 (m, 4B, B4, 6, 9, 10), −13.45 (s, 2B, B8, 11), −15.32 (s, 2B, B2, 3); UV (99.8% EtOH): λ_max_ = 257, 338 nm, λ_min_ = 295 nm, λ_sh_ = 233, 373 nm; FT-IR: *ν*_max_ (cm^−1^) = 2956 (C-H_aliphat_), 2593 (B-H), 1698 (C=O), 1660 (C=O), 730 (B-B); RP-HPLC (gradient B): *t*_R_ = 22.58 min; APCI-MS: *m*/*z*: 547 [M + H]^+^, calcd for C_25_H_35_B_10_N_5_O_2_ = 546.

#### 3.1.4. Synthesis of 1,8-Naphthalic Anhydride Derivatives **15**, **16**, and Naphthalimide Derivatives **17**–**20** Modified with Carborane Clusterc via Click Reactions

1-(3-Azidopropyl)-carborane (*ortho*-carborane (**4**) or *meta*-carborane (**5**)) (1 equiv.) was dissolved in THF/H_2_O (2.4:1, *v*/*v*, 2.5 mL per 0.1 mmoL). 3-(Prop-2-yn-1-yloxy)-1,8-naphthalic anhydride (**14**) (1 equiv.), CuSO_4_ 5H_2_O (0.06 equiv.) and sodium ascorbate (0.1 equiv.) were added. Reaction mixture was stirred for 3–4 h in 35 °C. For compound **15** after 1 h additional portion of CuSO_4_⋅5H_2_O (0.06 equiv.) and sodium ascorbate (0.1 equiv.) were added. The reaction was quenched by evaporation of the solvents. The crude compound was purified by column chromatography on silica gel (230–400 mesh) with a gradient of MeOH (0–5%) in CH_2_Cl_2_ as the eluent to afford product.

*3-{[1-(3-(1,2-Dicarba-closo-dodecaborane-1-yl)propyl)-1H-1,2,3-triazol-4-yl]methoxy}-1,8-naphthalic anhydride* (**15**): white solid. Yield: 93.6 mg (74%). TLC (MeOH/CH_2_Cl_2_, 1:49, *v*/*v*): *R*_f_ = 0.23; ^1^H-NMR (DMSO, 600.26 MHz): *δ* (ppm) = 8.41 (d, *J* = 7.7 Hz, 1H, H_arom_), 8.37 (dd, *J* = 7.2, 1.0 Hz, 1H, H_arom_), 8.31 (s, 1H, CH_triazole_), 8.17 (d, *J* = 2.6 Hz, 1H, H_arom_), 8.12 (d, *J* = 2.6 Hz, 1H, H_arom_), 7.87 (dd, *J* = 8.1, 7.3 Hz, 1H, H_arom_), 5.44 (s, 2H, O-CH_2_-triazole), 5.14 (br s, 1H, CH_carborane_), 4.37 (t, *J* = 6.9 Hz, 2H, CH_2_-triazole), 2.28–2.25 (m, 2H, CH_2_-carborane), 2.01–1.96 (m, 2H, CH_2_-CH_2_-CH_2_), 3.0–1.5 (m, 10H, B_10_H_10_); ^13^C-NMR (DMSO, 150.95 MHz): *δ* (ppm) = 160.48 (C11), 160.16 (C12), 156.37 (C3), 142.07 (C_triazole_), 134.05–115.85 (9C_arom_ + CH_triazole_), 75.42 (C_carborane_), 62.85 (CH_carborane_), 61.97 (O-CH_2_-triazole), 48.25 (CH_2_-triazole), 33.49 (CH_2_-carborane), 29.34 (CH_2_-CH_2_-CH_2_); ^11^B-NMR {1H BB} (DMSO, 192.59 MHz): *δ* (ppm) = −3.25 (s, 1B, B9), −6.23 (s, 1B, B12), −9.85 (s, 2B, B8,10), −11.76–−13.13 (m, 6B, B3,4,5,6,7,11); UV (99.8% EtOH): λ_max_ = 236.5, 329.1, 371.7 nm, λ_min_ = 227.5, 281.7, 344.9 nm, λ_sh_ = 245.9, 314.6 nm; FT-IR: ν_max_ (cm^−1^) = 2924 (C-H_aliphat_), 2579 (B-H), 1770 (C=O), 1732 (C=O), 724 (B-B); RP-HPLC (gradient A): *t*_R_ = 17.61 min; APCI-MS: *m*/*z*: 480 [M + H]^+^, 513 [M + H + MeOH]^+^, 522 [M + H + CH_3_CN]^+^, calcd for C_20_H_25_B_10_N_3_O_4_ = 479.

*3-{[1-(3-(1,7-Dicarba-closo-dodecaborane-1-yl)propyl)-1H-1,2,3-triazol-4-yl]methoxy}-1,8-naphthalic anhydride* (**16**): white solid. Yield: 39 mg (52%). TLC (MeOH/CH_2_Cl_2_, 1:49, *v*/*v*): *R*_f_ = 0.30; ^1^H-NMR (DMSO, 600.26 MHz): *δ* (ppm) = 8.41 (d, 1H, *J* = 8.3 Hz, H_arom_), 8.37 (d, 1H, *J* = 7.2 Hz, H_arom_), 8.30 (s, 1H, CH_triazole_), 8.17 (d, 1H, *J* = 2.4 Hz, H_arom_), 8.12 (d, 1H, *J* = 2.4 Hz, H_arom_), 7.87 (dd, 1H, *J* = 8.1, 7.5 Hz, H_arom_), 5.44 (s, 2H, O-CH_2_-triazole), 4.33 (t, 2H, *J* = 6.7 Hz, CH_2_-triazole), 4.01 (br s, 1H, CH_carborane_), 1.92–1.85 (m, 4H, CH_2_-carborane overlapped with CH_2_-C*H*_2_-CH_2_), 3.0–1.5 (m, 10H, B_10_H_10_); ^13^C-NMR (DMSO, 150.95 MHz): *δ* (ppm) = 160.52 (1C, C11), 160.18 (1C, C12), 156.35 (1C, C3), 142.03 (1C, C_triazole_), 134.07–115.87 (10C, 9C_arom_, CH_triazole_), 75.62 (1C, C_carborane_), 61.94 (1C, O-CH_2_-triazole), 56.26 (1C, CH_carborane_), 48.39 (1C, CH_2_-triazole), 32.72 (1C, CH_2_-carborane), 30.05 (1C, CH_2_-*C*H_2_-CH_2_); ^11^B-NMR {^1^H BB} (DMSO, 192.59 MHz): *δ* (ppm) = −4.56 (s, 1B, B5), −11.16 (s, 5B, B4, 6, 9, 10, 12), −13.61 (s, 2B, B8, 11), −15.15 (s, 2B, B2, 3); UV (99.8% EtOH): λ_max_ = 235, 328, 372 nm, λ_min_ = 227, 282, 346 nm, λ_sh_ = 314, 356 nm; FT-IR: *ν*_max_ (cm^−1^) = 2954 (C-H_aliphat_), 2596 (B-H), 1771 (C=O), 1733 (C=O), 725 (B-B); RP-HPLC (gradient A): *t*_R_ = 17.42 min; APCI-MS: *m*/*z*: 480 [M + H]^+^, 513 [M + H + MeOH]^+^, 522 [M + H + CH_3_CN]^+^, calcd for C_20_H_25_B_10_N_3_O_4_ = 479.

*3-{[1-(3-(1,2-Dicarba-closo-dodecaborane-1-yl)propyl)-1H-1,2,3-triazol-4-yl]methoxy}-1,8-naphthalic anhydride* (**15**) or *3-{[1-(3-(1,7-dicarba-closo-dodecaborane-1-yl) propyl)-1H-1,2,3-triazol-4-yl]methoxy}-1,8-naphthalic anhydride* (**16**) (1 equiv.) was dissolved in absolute EtOH (10 mL per 0.45 mmoL) and *N,N*-dimethylethylenediamine (1.1 equiv.) or *N*-(2-aminoethyl)pyrrolidine (1.1 equiv.) was added. The reaction mixture was stirred for 1 h at 35 °C, then for 1 h at 45 °C, under an inert (Ar) atmosphere. For compound **20** additional portion of *N*-(2-aminoethyl)pyrrolidine (0.6 equiv.) were added and reaction mixture was stirred for 2 h in 45 °C. Subsequently, the solvent was evaporated to dryness under vacuum and crude product was purified by column chromatography on silica gel (230–400 mesh) with a gradient of MeOH (0–15%) in CH_2_Cl_2_. Additionally (for compound **18** and **20**), purified product was dissolved in CHCl_3_ (0.5–1 mL) and poured into hexane (5–8 mL). A precipitate was isolated by centrifugation.

*N-[2-(Dimethylamino)ethyl]-3-{[1-(3-(1,2-dicarba-closo-dodecaborane-1-yl)propyl)-1H-1,2,3-triazol-4-yl]metoxy}-1,8-naphthalimide* (**17**): yellow solid. Yield: 14.4 mg (59%). TLC (MeOH/CH_2_Cl_2_, 1:9, *v*/*v*): *R*_f_ = 0.23; ^1^H-NMR (DMSO, 600.26 MHz): δ (ppm) = 8.32–8.30 (m, 3H, 2H_arom_ overlapped with CH_triazole_), 8.07 (d, 1H, J = 2.5 Hz, H_arom_), 8.05 (d, 1H, J = 2.5 Hz, H_arom_), 7.81 (t, 1H, J = 7.8 Hz, H_arom_), 5.41 (s, 2H, O-CH_2_-triazole), 5.13 (br s, 1H, CH_carborane_), 4.37 (t, 2H, J = 6.9 Hz, CH_2_-triazole), 4.15 (t, 2H, J = 6.9 Hz, CH_2_-N(CO)_2_), 2.57 (t, 2H, J = 6.7 Hz, CH_2_-N(CH_3_)_2_), 2.26–2.23 (m, 8H, 2N(CH_3_)_2_ overlapped with CH_2_-carborane), 2.00–1.95 (m, 2H, CH_2_-CH_2_-CH_2_), 3.0–1.5 (m, 10H, B_10_H_10_); ^13^C-NMR (DMSO, 150.95 MHz): δ (ppm) = 163.36 (1C, C11), 162.91 (1C, C12), 156.27 (1C, C3), 142.24 (1C, C_triazole_), 133.02–114.70 (10C, 9C_arom_, CH_triazole_), 75.41 (1C, C_carborane_), 62.84 (1C, CH_carborane_), 61.84 (1C, O-CH_2_-triazole), 56.26 (1C, CH_2_-N(CH_3_)_2_), 48.25 (1C, CH_2_-triazole), 45.10 (2C, 2 × CH_3_), 37.42 (1C, CH_2_-N(CO)_2_), 33.49 (1C, CH_2_-carborane), 29.35 (1C, CH_2_-CH_2_-CH_2_); ^11^B-NMR {^1^H BB} (DMSO, 192.59 MHz): δ (ppm) = −3.20 (s, 1B, B9), −6.15 (s, 1B, B12), −9.82 (s, 2B, B8,10), −11.72–−13.07 (m, 6B, B3, 4, 5, 6, 7, 11); UV (99.8% EtOH): λ_max_ = 238, 334, 375 nm, λ_min_ = 226, 280, 351 nm, λ_sh_ = 320 nm; FT-IR: ν_max_ (cm^−1^) = 2925 (C-H_aliphat_), 2594 (B-H), 1698 (C=O), 1658 (C=O), 723 (B-B); RP-HPLC (gradient B): t_R_ = 21.64 min; APCI-MS: *m*/*z*: 550 [M + H]^+^, calcd for C_24_H_35_B_10_N_5_O_3_ = 549.

*N-[2-(N-Pyrrolidinyl)ethyl]-3-{[1-(3-(1,2-dicarba-closo-dodecaborane-1-yl)propyl)-1H-1,2,3-triazol-4-yl]metoxy}-1,8-naphthalimide* (**18**): white solid. Yield: 22.7 mg (58%). TLC (MeOH/CH_2_Cl_2_, 1:9, *v*/*v*): *R*_f_ = 0.24; ^1^H-NMR (CDCl_3_, 600.26 MHz): δ (ppm) = 8.40 (d, 1H, J = 7.3 Hz, H_arom_), 8.22 (d, 1H, J = 2.6 Hz, H_arom_), 8.05 (d, 1H, J = 8.1 Hz, H_arom_), 7.69 (s, 1H, CH_triazole_), 7.67 (t, 1H, J = 7.5 Hz, H_arom_), 7.62 (d, 1H, J = 2.4 Hz, H_arom_), 5.39 (s, 2H, O-CH_2_-triazole), 4.38–4.35 (m, 4H, CH_2_-triazole overlapped with CH_2_-N(CO)_2_), 3.58 (s, 1H, CH_carborane_), 2.90 (t, 2H, J = 7.1 Hz, CH_2_-pyrrolidine), 2.77 (br s, 4H, N-CH_2pyrrolidine_-CH_2_), 2.25–2.22 (m, 2H, CH_2_-carborane), 2.16–2.13 (CH_2_-CH_2_-CH_2_), 1.83 (m, 2H, CH_2_-CH_2pyrrolidine_-CH_2_), 3.0–1.5 (m, 10H, B_10_H_10_); ^13^C-NMR (CDCl_3_, 150.95 MHz): δ (ppm) = 164.22 (1C, C11), 163.85 (1C, C12), 156.76 (1C, C3), 143.91 (1C, C_triazole_), 133.23–114.58 (10C, 9C_arom_, CH_triazole_), 73.77 (1C, C_carborane_), 62.68 (1C, O-CH_2_-triazole), 61.78 (1C, CH_carborane_), 54.50 (1C, N-CH_2pyrrolidine_-CH_2_), 53.71 (1C, CH_2_-pyrrolidine), 49.28 (1C, CH_2_-triazole), 38.99 (1C, CH_2_-N(CO)_2_), 35.08 (1C, CH_2_-carborane), 29.81 (1C, CH_2_-CH_2_-CH_2_), 23.77 (1C, CH_2_-CH_2pyrrolidine_-CH_2_); ^11^B-NMR {^1^H BB} (CDCl_3_, 192.59 MHz): δ (ppm) = −2.15 (s, 1B, B9), −5.45 (s, 1B, B12), −9.17 (s, 2B, B8, 10), −11.86–−12.97 (m, 6B, B3, 4, 5, 6, 7, 11); UV (99.8% EtOH): λ_max_ = 239, 335, 374 nm, λ_min_ = 225, 280, 352 nm, λ_sh_ = 318 nm; FT-IR: ν_max_ (cm^−1^) = 2926 (C-H_aliphat_), 2578 (B-H), 1698 (C=O), 1659 (C=O), 723 (B-B); RP-HPLC (gradient B): t_R_ = 21.87 min; APCI-MS: *m*/*z*: 577 [M + H]^+^, calcd for C_26_H_37_B_10_N_5_O_3_ = 576.

*N-[2-(Dimethylamino)ethyl]-3-{[1-(3-(1,7-dicarba-closo-dodecaborane-1-yl)propyl)-1H-1,2,3-triazol-4-yl]metoxy}-1,8-naphthalimide* (**19**): yellow solid. Yield: 16.3 mg (71%). TLC (MeOH/CH_2_Cl_2_, 1:9, *v*/*v*): *R*_f_ = 0.39; ^1^H-NMR (CDCl_3_, 600.26 MHz): δ (ppm) = 8.43 (d, 1H, J = 7.2 Hz, H_arom_), 8.26 (d, 1H, J = 2.0 Hz, H_arom_), 8.08 (d, 1H, J = 8.1 Hz, H_arom_), 7.69–7.64 (m, 3H, 2H_arom_ overlapped with CH_triazole_), 5.41 (s, 2H, O-CH_2_-triazole), 4.35 (t, 2H, J = 6.7 Hz, CH_2_-N(CO)_2_), 4.30 (t, 2H, J = 6.1 Hz, CH_2_-triazole), 2.90 (br s, 1H, CH_carborane_), 2.75 (br s, 2H, CH_2_-N(CH_3_)_2_), 2.42 (s, 6H, N(CH_3_)_2_), 1.98–1.97 (m, 4H, CH_2_-CH_2_-CH_2_ overlapped with CH_2_-carborane), 3.0–1.5 (m, 10H, B_10_H_10_); ^13^C-NMR (CDCl_3_, 150.95 MHz): δ (ppm) = 164.36 (1C, C11), 163.96 (1C, C12), 156.84 (1C, C3), 143.79 (1C, C_triazole_), 133.29–114.05 (10C, 9C_arom_, CH_triazole_), 74.65 (1C, C_carborane_), 62.72 (1C, O-CH_2_-triazole), 56.91 (1C, CH_2_-N(CH_3_)_2_), 55.19 (1C, CH_carborane_), 49.64 (1C, CH_2_-triazole), 45.56 (2C, 2 × CH_3_), 37.95 (1C, CH_2_-N(CO)_2_), 33.75 (1C, CH_2_-carborane), 30.45 (1C, CH_2_-CH_2_-CH_2_); ^11^B-NMR {^1^H BB} (CDCl_3_, 192.59 MHz): δ (ppm) = −4.26 (s, 1B, B5), −9.68 (s, 1B, B12), −10.68–−11.06 (m, 4B, B4, 6, 9, 10), −13.49 (s, 2B, B8, 11), −15.38 (s, 2B, B2, 3); UV (99.8% EtOH): λ_max_ = 238, 334, 375 nm, λ_min_ = 225, 283, 351 nm, λ_sh_ = 319 nm; FT-IR: ν_max_ (cm^−1^) = 2936 (C-H_aliphat_), 2595 (B-H), 1698 (C=O), 1655 (C=O), 731 (B-B); RP-HPLC (gradient B): t_R_ = 21.23 min; APCI-MS: *m*/*z*: 550 [M + H]^+^, calcd for C_24_H_35_B_10_N_5_O_3_ = 549.

*N-[2-(N-Pyrrolidinyl)ethyl]-3-{[1-(3-(1,7-dicarba-closo-dodecaborane-1-yl)propyl)-1H-1,2,3-triazol-4-yl]metoxy}-1,8-naphthalimide* (**20**): white solid. Yield: 28.3 mg (83%). TLC (MeOH/CH_2_Cl_2_, 1:9, *v*/*v*): *R*_f_ = 0.39; ^1^H-NMR (CDCl_3_, 600.26 MHz): δ (ppm) = 8.40 (d, 1H, J = 7.2 Hz, H_arom_), 8.23 (d, 1H, J = 2.3 Hz, H_arom_), 8.05 (d, 1H, J = 8.1 Hz, H_arom_), 7.66–7.63 (m, 3H, 2H_arom_ overlapped with CH_triazole_), 5.40 (s, 2H, O-CH_2_-triazole), 4.34 (t, 2H, J = 7.2 Hz, CH_2_-N(CO)_2_), 4.30 (t, 2H, J = 6.4 Hz, CH_2_-triazole), 2.90 (br s, 1H, CH_carborane_), 2.83 (t, 2H, J = 7.2 Hz, CH_2_-pyrrolidine), 2.70 (br s, 4H, N-CH_2pyrrolidine_-CH_2_), 2.01–1.95 (m, 4H, CH_2_-CH_2_-CH_2_ overlapped with CH_2_-carborane), 1.80 (br s, 4H, CH_2_-CH_2pyrrolidine_-CH_2_), 3.0–1.5 (m, 10H, B_10_H_10_); ^13^C-NMR (CDCl_3_, 150.95 MHz): δ (ppm) = 164.22 (1C, C11), 163.80 (1C, C12), 156.81 (1C, C3), 143.76 (1C, C_triazole_), 133.24–114.33 (10C, 9C_arom_, CH_triazole_), 74.64 (1C, C_carborane_), 62.67 (1C, O-CH_2_-triazole), 55.18 (1C, CH_carborane_), 54.47 (1C, N-CH_2pyrrolidine_-CH_2_), 53.75 (1C, CH_2_-pyrrolidine), 49.61 (1C, CH_2_-triazole), 39.25 (1C, CH_2_-N(CO)_2_), 33.73 (1C, CH_2_-carborane), 30.43 (1C, CH_2_-CH_2_-CH_2_), 23.77 (1C, CH_2_-CH_2pyrrolidine_-CH_2_); ^11^B-NMR {^1^H BB} (CDCl_3_, 192.59 MHz): δ (ppm) = −4.24 (s, 1B, B5), −9.68 (s, 1B, B12), −10.64–−11.06 (m, 4B, B4, 6, 9, 10), −13.49 (s, 2B, B8, 11), −15.38 (s, 2B, B2, 3); UV (99.8% EtOH): λ_max_ = 238, 334, 375 nm, λ_min_ = 226, 286, 351 nm, λ_sh_ = 319 nm; FT-IR: ν_max_ (cm^−1^) = 2950 (C-H_aliphat_), 2595 (B-H), 1698 (C=O), 1658 (C=O), 731 (B-B); RP-HPLC (gradient B): t_R_ = 21.32 min; APCI-MS: *m*/*z*: 577 [M+H]^+^, calcd for C_26_H_37_B_10_N_5_O_3_ = 576; HRMS (ESI+) 576.4156 [M + H]^+^, calcd for C_26_H_37_B_10_N_5_O_3_ = 576.3973 [M + H]^+^.

#### 3.1.5. Synthesis of 1,8-Naphthalic Anhydride Derivatives **31**, **32** Modified with Carborane Cluster via Reductive Amination

*3-Amino-1,8-naphthalic anhydride* (**12**) (10–12.9 mg, 46.9–60.6 μmoL) was dissolved in anhydrous THF (0.9 mL) and *2-(1,2-dicarba-closo-dodecaborae-1-yl)ethanal* (**23**) (1.3 equiv.) or *2-(1,7-dicarba-closo-dodecaborae-1-yl)ethanal* (**24**) (1.2 equiv.) was added. The reaction mixture was stirred for 24 h at reflux under an inert (Ar) atmosphere. Next, to the Schiff base **25**, **26** NaBH_3_CN (3 equiv.) was added and reaction mixture was stirred for 24 h at RT under an inert (Ar) atmosphere. Subsequently, concentrated HCl (4–6 equiv.) was added and the reaction was stirred for additional 1 h. Then the reaction mixture was diluted with brine (2–4 mL) and crude product was extracted with CH_2_Cl_2_ (3 × 2–5 mL). The organic phase was separated, dried over MgSO_4_, filtered, and evaporated to dryness. The residue was purified by column chromatography on silica gel (230–400 mesh) using as an eluent CH_2_Cl_2_ to afford product.

*3-[(1,2-Dicarba-closo-dodecaborane-1-yl)ethylamino]-1,8-naphthalic anhydride* (**31**): yellow solid. Yield: 16.3 mg (62%). TLC (2 × CH_2_Cl_2_): *R*_f_ = 0.33. ^1^H-NMR (DMSO-d_6_, 600.26 MHz): *δ* (ppm) = 8.18 (d, 1H, *J* = 8.3 Hz, H_arom_), 8.11 (d, 1H, *J* = 7.2 Hz, H_arom_), 7.94 (d, 1H, *J* = 2.4 Hz, H_arom_), 7.69 (t, 1H, *J* = 7.7 Hz, H_arom_), 7.30 (d, 1H, *J* = 2.3 Hz, H_arom_), 6.67 (t, 1H, *J* = 5.5 Hz, NH), 5.26 (br s, 1H, CH_carborane_), 3.32 (signal of C*H*_2_-NH overlapped with H_2_O), 2.62 (t, 2H, *J* = 7.5 Hz, CH_2_-carborane), 3.0–1.5 (m, 10H, B_10_H_10_); ^13^C-NMR (DMSO-d_6_, 150.95 MHz): *δ* (ppm) = 160.88 (1C, C11), 160.79 (1C, C12), 146.75–109.25 (10C, 10C_arom_), 74.28 (1C, C_carborane_), 62.94 (1C, CH_carborane_), 41.79 (1C, CH_2_-NH), 34.94 (1C, CH_2_-carborane); ^11^B-NMR {^1^H BB} (CDCl_3_, 192.59 MHz): *δ* (ppm) = −3.05 (s, 1B, B9), −5.91 (s, 1B, B12), −9.76 (s, 2B, B8,10), −11.45–−13.06 (m, 6B, B3, 4, 5, 6, 7, 11); UV (99.8% EtOH): λ_max_ = 277, 336, 437 nm, λ_min_ = 239, 305, 359 nm; FT-IR: *ν*_max_ (cm^−1^) = 2926 (C-H_aliphat_), 2587 (B-H), 1763 (C=O), 1733 (C=O), 724 (B-B); RP-HPLC (gradient A): *t*_R_ = 18.50 min; APCI-MS: *m*/*z*: 384 [M + H]^+^, 416 [M+MeOH+H]^+^calcd for C_16_H_21_B_10_NO_3_ = 383.

*3-[(1,7-Dicarba-closo-dodecaborane-1-yl)ethylamino]-1,8-naphthalic anhydride* (**32**): yellow solid. Yield: 9.8 mg (55%). TLC (2 × CH_2_Cl_2_): *R*_f_ = 0.38. ^1^H-NMR (DMSO-d_6_, 600.26 MHz): *δ* (ppm) = 8.19 (d, 1H, *J* = 7.8 Hz, H_arom_), 8.11 (dd, 1H, *J* = 7.2, 0.9 Hz, H_arom_) 7.93 (d, 1H, *J* = 2.4 Hz, H_arom_), 7.69 (dd, 1H, *J* = 8.2, 7.3 Hz, H_arom_), 7.27 (d, 1H, *J* = 2.3 Hz, H_arom_), 6.61 (t, 1H, *J* = 5.5 Hz, NH), 4.08 (br s, 1H, CH_carborane_), 3.21 (dd, 2H, *J* = 13.1, 7.3 Hz, C*H*_2_-NH), 2.31 (t, 2H, *J* = 7.5 Hz, CH_2_-carborane), 3.0–1.5 (m, 10H, B_10_H_10_); ^13^C-NMR (DMSO-d_6_, 150.95 MHz): *δ* (ppm) = 160.92 (1C, C11), 160.82 (1C, C12), 146.87–109.09 (10C, 10C_arom_), 74.33 (1C, C_carborane_), 56.48 (1C, CH_carborane_), 42.29 (1C, CH_2_-NH), 34.38 (1C, CH_2_-carborane); ^11^B-NMR {^1^H BB} (DMSO-d_6_, 192.59MHz): *δ* (ppm) = −4.59 (s, 1B, B5), −11.11 (s, 5B, B4, 6, 9, 10, 12), −13.61–−15.04 (m, 4B, B2, 3, 8, 11); UV (99.8% EtOH): λ_max_ = 279, 336, 439 nm, λ_min_ = 239, 305, 360 nm; FT-IR: *ν*_max_ (cm^−1^) = 2926 (C-H_aliphat_), 2596 (B-H), 1762 (C=O), 1732 (C=O), 724 (B-B); RP-HPLC (gradient A): *t*_R_ = 19.17 min; APCI-MS: *m*/*z*: 384 [M + H]^+^, 416 [M+MeOH+H]^+^calcd for C_16_H_21_B_10_NO_3_ = 383.

#### 3.1.6. Synthesis of Naphthalimide Derivatives **33**-**36** Modified with Carborane Cluster via Reductive Amination

*3-Amino-N-[2-(dimethylamino)ethyl]-1,8-naphthalimide* (**21**) (10–17.2 mg, 35.3–60.7 µmoL) or *3-amino-N-[2-(N-pyrrolidinyl)ethyl]-1,8-naphthalimide* (**22**) (11–32 mg, 35.6–103.6 µmoL) was dissolved in anhydrous THF (14.9 mL per 1 mmoL) (for synthesis of compound **33**, **34**) or in anhydrous MeOH (14 mL per 1 mmoL) (for synthesis of compound **35**, **36**) and 2-(1,2-dicarba-*closo*-dodecaborae-1-yl)ethanal (**23**) (1.3 equiv.) or 2-(1,7-dicarba-*closo*-dodecaborae-1-yl)ethanal (**24**) (1.3 equiv.) was added. The reaction mixture was stirred for 24 h at reflux under an inert (Ar) atmosphere. Next, to the Schiff base **27**–**30** NaBH_3_CN (3 equiv.) was added and reaction mixture was stirred for 24 h at RT under an inert (Ar) atmosphere. Subsequently, concentrated HCl (5–6 equv.) was added and the reaction was stirred for additional 1 h. Then, the reaction mixture was diluted with brine (57 mL per 1 mmoL) and crude product was extracted with CH_2_Cl_2_ (4 × 2–10 mL). The organic phase was separated, dried over MgSO_4_, filtered, and evaporated to dryness. Crude product was purified by column chromatography on silica gel (230–400 mesh) with a gradient of MeOH (3–14%) in CH_2_Cl_2_ as the eluent to afford product.

*N-[2-(Dimethylamino)ethyl]-3-[1,2-dicarba-closo-dodecaborane-1-yl)ethylamino]-1,8-naphthalimide* (**33**): yellow solid. Yield: 15.7 mg (57%). TLC (MeOH/CH_2_Cl_2_, 1:9, *v*/*v*): *R*_f_ = 0.30. ^1^H-NMR (DMSO-d_6_, 600.26 MHz): *δ* (ppm) = 8.11–8.08 (m, 2H, 2H_arom_), 7.95 (d, 1H, *J* = 2.4 Hz, H_arom_), 7.65 (dd, 1H, *J* = 8.1, 7.3 Hz, H_arom_), 7.23 (d, 1H, *J* = 2.4 Hz, H_arom_), 6.58 (t, 1H, *J* = 5.7 Hz, NH), 5.27 (br s, 1H, CH_carborane_), 4.12 (t, 2H, *J* = 6.9 Hz, CH_2_-N(CO)_2_), 3.31 (signal of C*H*_2_-NH overlapped with H_2_O), 2.62 (t, 2H, *J* = 7.5 Hz, CH_2_-carborane), 2.50 (signal of C*H*_2_-N(CH_3_)_2_ overlapped with DMSO), 2.21 (s, 6H, N(CH_3_)_2_), 3.0–1.5 (m, 10H, B_10_H_10_); ^13^C-NMR (DMSO-d_6_, 150.95 MHz): *δ* (ppm) = 163.68 (1C, C11), 163.47 (1C, C12), 146.55–108.34 (10C, 10C_arom_), 74.33 (1C, C_carborane_), 62.94 (1C, CH_carborane_), 56.45 (1C, *C*H_2_-N(CH_3_)_2_), 45.30 (2C, 2 × CH_3_), 41.83 (1C, CH_2_-NH), 37.45, (1C, *C*H_2_-N(CO)_2_), 34.99 (CH_2carborane_); ^11^B-NMR {^1^H BB} (DMSO-d_6_, 192.59MHz): *δ* (ppm) = −3.17 (s, 1B, B9), −6.02 (s, 1B, B12), −9.82 (s, 2B, B8, 10), −11.62–−13.11 (m, 6B, B3, 4, 5, 6, 7, 11); UV (99.8% EtOH): λ_max_ = 256, 341, 438 nm, λ_min_ = 236, 312, 366 nm, λ_sh_ = 280 nm; FT-IR: *ν*_max_ (cm^−1^) = 2973 (C-H_aliphat_), 2579 (B-H), 1704 (C=O),1651 (C=O), 721 (B-B); RP-HPLC (gradient B): *t*_R_ = 22.36 min; APCI-MS: *m*/*z*: 454 [M + H]^+^, calcd for C_20_H_31_B_10_N_3_O_2_ = 453.

*N-[2-(N-Pyrrolidinyl)ethyl]-3-[1,2-dicarba-closo-dodecaborane-1-yl)ethylamino]-1,8-naphthalimide* (**34**): yellow solid. Yield: 23.5 mg (47%). TLC (MeOH/CH_2_Cl_2_, 1:9, *v*/*v*): *R*_f_ = 0.30. ^1^H-NMR (DMSO-d_6_, 600.26 MHz): *δ* (ppm) = 8.12–8.09 (m, 2H, 2H_arom_), 7.96 (d, 1H, *J* = 2.4 Hz, H_arom_), 7.65 (t, 1H, *J* = 7.8 Hz, H_arom_), 7.24 (d, 1H, *J* = 2.3 Hz, H_arom_), 6.59 (t, 1H, *J* = 5.6 Hz, NH), 5.26 (br s, 1H, CH_carborane_), 4.17 (t, 2H, *J* = 6.9 Hz, CH_2_-N(CO)_2_), 3.31 (signal of C*H*_2_-NH overlapped with H_2_O), 2.78 (br s, 2H, CH_2_-pyrrolidine), 2.65–2.61 (m, 6H, N-C*H*_2pyrrolidine_-CH_2_ overlapped with CH_2_-carborane), 1.70 (br s, 4H, CH_2_-C*H*_2pyrrolidine_-CH_2_), 3.0–1.5 (m, 10H, B_10_H_10_); ^13^C-NMR (DMSO-d_6_, 150.95 MHz): *δ* (ppm) = 163.77 (1C, C11), 163.55 (1C, C12), 146.57–108.37 (10C, 10C_arom_), 74.35 (1C, C_carborane_), 62.95 (1C, CH_carborane_), 53.75 (1C, N-*C*H_2pyrrolidine_-CH_2_), 52.92 (1C, CH_2_-pyrrolidine), 41.83 (1C, CH_2_-NH), 38.21 (1C, *C*H_2_-N(CO)_2_), 34.99 (1C, CH_2_-carborane), 23.11 (1C, CH_2_-*C*H_2pyrrolidine_-CH_2_); ^11^B-NMR {^1^H BB} (DMSO-d_6_, 192.59MHz): *δ* (ppm) = −3.14 (s, 1B, B9), −6.10 (s, 1B, B12), −9.82–−13.12 (m, 8B, B3, 4, 5, 6, 7, 8, 10, 11); UV (99.8% EtOH): λ_max_ = 25, 341, 439 nm, λ_min_ = 236, 314.4, 366 nm, λ_sh_ = 280 nm; FT-IR: *ν*_max_ (cm^−1^) = 2956 (C-H_aliphat_), 2576 (B-H), 1697 (C=O), 1656 (C=O), 722 (B-B), RP-HPLC (gradient B) *t*_R_ = 22.83 min; APCI-MS: *m*/*z*: 480 [M + H]^+^, calcd for C_22_H_33_B_10_N_3_O_2_ = 479.

*N-[2-(Dimethylamino)ethyl]-3-[1,7-dicarba-closo-dodecaborane-1-yl)ethylamino]-1,8-naphthalimide* (**35**): yellow solid. Yield: 11.7 mg (73%). TLC (MeOH/CH_2_Cl_2_, 1:9, *v*/*v*): *R*_f_ = 0.33. ^1^H-NMR (CDCl_3_, 600.26 MHz): *δ* (ppm) = 8.28 (dd, 1H, *J* = 7.2, 0.6 Hz, H_arom_), 7.92 (d, 1H, *J* = 8.1 Hz, H_arom_), 7.88 (d, 1H, *J* = 2.4 Hz, H_arom_), 7.60 (t, 1H, *J* = 7.7 Hz, H_arom_), 6.98 (d, 1H, *J* = 2.3 Hz, H_arom_), 4.35 (t, 2H, *J* = 7.0 Hz, CH_2_-N(CO)_2_), 4.24 (t, 1H, *J* = 5.7 Hz, NH), 3.31 (dd, 2H, *J* = 13.6, 7.2 Hz, C*H*_2_-NH), 2.98 (br s, 1H, CH_carborane_), 2.75 (t, 2H, *J* = 6.7 Hz, C*H*_2_-N(CH_3_)_2_), 2.43 (s, 6H, N(CH_3_)_2_), 2.33 (t, 2H, *J* = 7.5 Hz, CH_2_-carborane), 3.0–1.5 (m, 10H, B_10_H_10_); ^13^C-NMR (CDCl_3_, 150.95 MHz): *δ* (ppm) = 164.65 (1C, C11), 164.41 (1C, C12), 145.79–109.83 (10C, 10C_arom_), 73.48 (1C, C_carborane_), 57.02 (1C, *C*H_2_-N(CH_3_)_2_), 55.37 (1C, CH_carborane_), 45.66 (2C, 2 × CH_3_), 43.24 (1C, CH_2_-NH), 37.88, (1C, *C*H_2_-N(CO)_2_), 35.52 (1C, CH_2_-carborane); ^11^B-NMR {^1^H BB} (CDCl_3_, 192.59MHz): *δ* (ppm) = −4.14 (s, 1B, B5), −9.43–−10.87 (m, 5B, B4, 6, 9, 10, 12), −13.37 (s, 2B, B8, 11), −15.19 (s, 2B, B2, 3); UV (99.8% EtOH): λ_max_ = 257, 341, 440 nm, λ_min_ = 236, 314, 367 nm, λ_sh_ = 280 nm; FT-IR: *ν*_max_ (cm^−1^) = 2953 (C-H_aliphat_), 2598 (B-H), 1703 (C=O), 1655 (C=O), 729 (B-B); RP-HPLC O (gradient B): *t*_R_ = 22.52 min; APCI-MS: *m*/*z*: 454 [M+H]^+^, calcd for C_20_H_31_B_10_N_3_O_2_ = 453; HRMS (ESI+) 454.3642 [M + H]^+^, calcd for C_20_H_31_B_10_N_3_O_2_ = 454.3493 [M + H]^+^.

*N-[2-(N-Pyrrolidinyl)ethyl]-3-[1,7-dicarba-closo-dodecaborane-1-yl)ethylamino]-1,8-naphthalimide* (**36**): yellow solid. Yield: 12.5 mg (74%). TLC (MeOH/CH_2_Cl_2_, 1:9, *v*/*v*): *R*_f_ = 0.33. ^1^H-NMR (CDCl_3_, 600.26 MHz): *δ* (ppm) = 8.27 (d, 1H, *J* = 7.3 Hz, H_arom_), 7.91 (d, 1H, *J* = 8.1 Hz, H_arom_), 7.87 (d, 1H, *J* = 2.3 Hz, H_arom_), 7.59 (t, 1H, *J* = 7.7 Hz, H_arom_), 6.95 (d, 1H, *J* = 2.3 Hz, H_arom_), 4.36 (t, 2H, *J* = 7.2 Hz, CH_2_-N(CO)_2_), 4.27 (t, 1H, *J* = 5.7 Hz, NH), 3.30 (dd, 2H, *J* = 13.8, 7.2 Hz, C*H*_2_-NH), 2.98 (br s, 1H, CH_carborane_), 2.86 (t, 2H, *J* = 7.0 Hz, CH_2_-pyrrolidine), 2.73 (br s, 4H, N-C*H*_2pyrrolidine_-CH_2_), 2.32 (t, 2H, *J* = 7.5 Hz, CH_2_-carborane), 1.82 (br s, 4H, CH_2_-C*H*_2pyrrolidine_-CH_2_), 3.0–1.5 (m, 10H, B_10_H_10_); ^13^C-NMR (CDCl_3_, 150.95 MHz): *δ* (ppm) = 164.58 (1C, C11), 164.32 (1C, C12), 145.78–109.69 (10C, 10C_arom_), 73.47 (1C, C_carborane_), 55.36 (1C, CH_carborane_), 54.50 (1C, N-*C*H_2pyrrolidine_-CH_2_), 53.87 (1C, CH_2_-pyrrolidine), 43.23 (1C, CH_2_-NH), 39.11, (1C, CH_2_-N(CO)_2_), 35.48 (1C, CH_2_-carborane), 23.79 (1C, CH_2_-*C*H_2pyrrolidine_-CH_2_); ^11^B-NMR {^1^H BB} (CDCl_3_, 192.59MHz): *δ* (ppm) = −4.18 (s, 1B, B5), −9.40 (s, 1B, B12), −10.52–−10.92 (m, 4B, B4, 6, 9, 10), −13.41 (s, 2B, B8, 11), −15.22 (s, 2B, B2, 3); UV (99.8% EtOH): λ_max_ = 257, 341, 441 nm, λ_min_ = 236, 313, 367 nm, λ_sh_ = 280 nm; FT-IR: *ν*_max_ (cm^−1^) = 2961 (C-H_aliphat_), 2593 (B- H), 1697 (C=O),1657 (C=O), 728 (B-B); RP-HPLC (gradient B): *t*_R_ = 22.95 min; APCI-MS: *m*/*z*: 480 [M + H]^+^, calcd for C_22_H_33_B_10_N_3_O_2_ = 479.

#### 3.1.7. Synthesis of Naphthalimide Derivatives **39**-**42** Modified with Carborane Cluster via Amidation Reaction

*3-Amino-N-[2-(dimethylamino)ethyl]-1,8-naphthalimide* (**21**) (29.4 mg, 104 μmoL) or *3-amino-N-[2-(N-pyrrolidinyl)ethyl]-1,8-naphthalimide* (**22**) (32.1 mg, 104 μmoL) and *3-(1,2-dicarba-closo-dodecaboran-1-yl)propionic acid* (**37**) (15 mg, 69.3 μmoL) or *3-(1,7-dicarba-closo-dodecaboran-1-yl)propionic acid* (**38**) (15 mg, 69.3 μmoL) were dissolved in anhydrous CH_2_Cl_2_ (0.6 mL), then anhydrous TEA (19.3 μL, 138.6 μmoL) and PyBOP (36 mg, 69.3 μmoL) were added. The reaction mixture was stirred for 2–3 h at RT under an inert (Ar) atmosphere. After that an additional portion of PyBOP (18 mg, 34.7 μmoL) was added and mixture was stirred for 2–3 h at RT under an inert (Ar) atmosphere. The reaction mixture was diluted with CH_2_Cl_2_ (2 mL) and washed witH-NaHCO_3_ (5% *w*/*v*, 6 × 1.5 mL) and water (2 × 1.5 mL). The organic layer was dried over MgSO_4_, then the drying agent was filtered off and washed with CH_2_Cl_2_. Filtrate and washings were combined, and evaporated to dryness under a vacuum. The crude product was purified twice by column chromatography on silica gel (230–400 mesh) using a gradient of MeOH (3–10%) in CH_2_Cl_2_ as an eluting solvent system.

*N-[2-(Dimethylamino)ethyl]-3-[(1,2-dicarba-closo-dodecaborane-1-yl)propanamido]-1,8-naphthalimide* (**39**): white solid. Yield: 18 mg (54%). TLC (MeOH/CH_2_Cl_2_, 1:9, *v*/*v*): *R*_f_ = 0.31. ^1^H-NMR (acetone-d_6_, 600.26 MHz): *δ* (ppm) = 9.85 (s, 1H, NH), 8.69 (d, 1H, *J* = 2.1 Hz, H_arom_), 8.56 (d, 1H, *J* = 2.1 Hz, H_arom_), 8.38 (d, 1H, *J* = 7.2 Hz, H_arom_), 8.24 (d, 1H, *J* = 8.3 Hz, H_arom_), 7.77 (t, 1H, *J* = 7.7 Hz, H_arom_), 4.75 (br s, 1H, CH_carborane_), 4.24 (t, 2H, *J* = 7.0 Hz, CH_2_-N(CO)_2_), 2.84 (br s, 4H, C*H*_2_-CONH overlapped with CH_2_-carborane), 2.59 (t, *J* = 7.0 Hz, 2H, C*H*_2_-N(CH_3_)_2_), 2.27 (s, 6H, N(CH_3_)_2_), 3.0–1.5 (m, 10H, B_10_H_10_); ^13^C-NMR (acetone-d_6_, 150.95 MHz): *δ* (ppm) = 170.29 (1C, C_amide_), 164.50 (1C, C11), 164.22 (1C, C12), 138.74–121.91 (10C, 10C_arom_), 76.53 (1C, C_carborane_), 63.88 (1C, CH_carborane_), 57.74 (1C, *C*H_2_-N(CH_3_)_2_), 46.01 (2C, 2 × CH_3_), 38.81 (1C, *C*H_2_-N(CO)_2_), 36.72 (1C, C*H*_2_-CONH). 33.37 (1C, CH_2_-carborane); ^11^B-NMR {^1^H BB} (acetone-d_6_, 192.59MHz): *δ* (ppm) = −2.69 (s, 1B, B9), −5.98 (s, 1B, B12), −9.67 (s, 2B, B8, 10), −11.55–−12.92 (s, 6B, B3, 4, 5, 6, 7, 11); UV (99.8% EtOH): λ_max_ = 253, 337, 375 nm, λ_min_ = 293, 357 nm, λ_sh_ = 228 nm; FT-IR: *ν*_max_ (cm^−1^) = 2923 (C-H_aliphat_), 2581 (B-H), 1698 (C=O), 1656 (C=O), 722 (B-B); RP-HPLC (gradient B): *t*_R_ = 21.35 min; APCI-MS: *m*/*z*: 482 [M+H]^+^, calcd for C_21_H_31_B_10_N_3_O_3_ = 481.

*N-[2-(N-Pyrrolidinyl)ethyl]-3-[(1,2-dicarba-closo-dodecaborane-1-yl)propanamido]-1,8-naphthalimide* (**40**): white solid. Yield: 23.2 mg (66%). TLC (MeOH/CH_2_Cl_2_, 1:9, *v*/*v*): *R*_f_ = 0.31. ^1^H-NMR (DMSO-d_6_, 600.26 MHz): *δ* (ppm) = 10.64 (s, 1H, NH), 8.67 (d, 1H, *J* = 1.9 Hz, H_arom_), 8.60 (d, 1H, *J* = 1.9 Hz, H_arom_), 8.35–8.34 (m, 2H, 2H_arom_), 7.79 (t, 1H, *J* = 7.7 Hz, H_arom_), 5.24 (br s, 1H, CH_carborane_), 4.18 (t, 2H, *J* = 7.0 Hz, CH_2_-N(CO)_2_), 2.76 (br s, 2H, C*H*_2_- pyrrolidine), 2.71–2.67 (m, 4H, C*H*_2_-CONH overlapped with CH_2_-carborane), 2.62 (br s, 4H, N-C*H*_2pyrrolidine_-CH_2_), 1.69 (br s, 4H, CH_2_-C*H*_2pyrrolidine_-CH_2_), 3.0–1.5 (m, 10H, B_10_H_10_); ^13^C-NMR (DMSO-d_6_, 150.95 MHz): *δ* (ppm) = 169.32 (1C, C_amide_), 163.32 (1C, C_imide_), 163.10 (1C, C_imide_), 137.65–120.79 (10C, 10C_arom_), 75.66 (1C, C_carborane_), 63.47 (1C, CH_carborane_), 53.69 (1C, N-*C*H_2pyrrolidine_-CH_2_), 52.85 (1C, CH_2_-pyrrolidine), 38.42 (1C, *C*H_2_-N(CO)_2_), 35.59 (1C, *C*H_2_-CONH). 31.88 (1C, CH_2_-carborane), 23.11 (1C, CH_2_-*C*H_2pyrrolidine_-CH_2_); ^11^B-NMR {^1^H BB} (DMSO-d_6_, 192.59 MHz): *δ* (ppm) = −3.10 (s, 1B, B9), −6.05 (s, 1B, B12), −9.75 (s, 2B, B8, 10), −11.69–−12.92 (m, 6B, B3, 4, 5, 6, 7, 11); UV (99.8% EtOH): λ_max_ = 254, 338, 375 nm, λ_min_ = 293, 357 nm, λ_sh_ = 228 nm; FT-IR: *ν*_max_ (cm^−1^) = 2922 (C-H_aliphat_), 2576 (B-H), 1704 (C=O), 1661 (C=O), 722 (B-B); RP-HPLC (gradient B): *t*_R_ = 21.39 min; APCI-MS: *m*/*z*: 508 [M+H]^+^, calcd for C_23_H_33_B_10_N_3_O_3_ = 507; HRMS (ESI+) 508.3761 [M+H]^+^, calcd for C_23_H_33_B_10_N_3_O_3_ = 508.3598 [M + H]^+^.

*N-[2-(Dimethylamino)ethyl]-3-[(1,7-dicarba-closo-dodecaborane-1-yl)propanamido]-1,8-naphthalimide* (**41**): white solid. Yield: 18.2 mg, (55%). TLC (MeOH/CH_2_Cl_2_, 1:9, *v*/*v*): *R*_f_ = 0.34. ^1^H-NMR (acetone-d_6_, 600.26 MHz): *δ* (ppm) = 9.72 (s, 1H, NH), 8.73 (d, 1H, *J* = 2.1 Hz, H_arom_), 8.55 (d, 1H, *J* = 2.1 Hz, H_arom_), 8.39 (dd, 1H, *J* = 7.2, 1.0 Hz, H_arom_), 8.26 (d, 1H, *J* = 8.3 Hz, H_arom_), 7.79 (dd, 1H, *J* = 8.1, 7.3 Hz, H_arom_), 4.25 (t, 2H, *J* = 6.9 Hz, CH_2_-N(CO)_2_), 3.71 (br s, 1H, CH_carborane_), 2.67 (dd, 2H, *J* = 9.3, 6.4 Hz, C*H*_2_-CONH), 2.61 (t, 2H, *J* = 7.0 Hz, C*H*_2_-N(CH_3_)_2_), 2.54 (dd, 2H, *J* = 9.3, 6.7 Hz, CH_2_-carborane), 2.30 (s, 6H, N(CH_3_)_2_), 3.0–1.5 (m, 10H, B_10_H_10_); ^13^C-NMR (acetone-d_6_, 150.95 MHz): *δ* (ppm) = 170.34 (1C, C_amide_), 164.46 (1C, C_imide_), 164.18 (1C, C_imide_), 138.85–121.73 (10C, 10C_arom_), 76.72 (1C, C_carborane_), 57.74 (1C, *C*H_2_-N(CH_3_)_2_), 56.92 (1C, CH_carborane_), 46.03 (2C, 2 × CH_3_), 38.79 (1C, *C*H_2_-N(CO)_2_), 37.14 (1C, C*H*_2_-CONH), 32.49 (1C, CH_2_-carborane); ^11^B-NMR {^1^H BB} (acetone-d_6_, 192.59 MHz): *δ* (ppm) = −4.33 (s, 1B, B5), −9.87–−10.92 (m, 5B, B4, 6, 9, 10, 12), −13.45 (s, 2B, B8, 11), −14.93 (s, 2B, B2, 3); UV (99.8% EtOH): λ_max_ = 258, 338, 375 nm, λ_min_ = 293, 358 nm, λ_sh_ = 228 nm; FT-IR: *ν*_max_ (cm^−1^) = 2922 (C-H_aliphat_), 2597 (B-H), 1697 (C=O), 1657 (C=O), 730 (B-B); RP-HPLC (gradient B): *t*_R_ = 21.03 min; APCI-MS: *m*/*z*: 482 [M + H]^+^, calcd for C_21_H_31_B_10_N_3_O_3_ = 481.

*N-[2-(N-Pyrrolidinyl)ethyl]-3-[(1,7-dicarba-closo-dodecaborane-1-yl)propanamido]-1,8-naphthalimide* (**42**): white solid. Yield: 17.2 mg (49%). TLC (MeOH/CH_2_Cl_2_, 1:9, *v*/*v*): *R*_f_ = 0.34. ^1^H-NMR (DMSO-d_6_, 600.26 MHz): *δ* (ppm) = 10.59 (s, 1H, NH), 8.69 (d, 1H, *J* = 1.9 Hz, H_arom_), 8.58 (d, 1H, *J* = 2.1 Hz, H_arom_), 8.35–8.33 (m, 2H, 2H_arom_), 7.79 (t, 1H, *J* = 7.7 Hz, H_arom_), 4.18 (t, 2H, *J* = 7.1 Hz, CH_2_-N(CO)_2_), 4.09 (br s, 1H, CH_carborane_), 2.72 (br s, 2H, C*H*_2_-pyrrolidine), 2.59 (br s, 4H, N-C*H*_2pyrrolidine_-CH_2_), 2.54 (t, 2H, *J* = 8.6 Hz, C*H*_2_-CONH), 2.40 (t, 2H, *J* = 8.0 Hz, CH_2_-carborane), 1.69 (br s, 4H, CH_2_-C*H*_2pyrrolidine_-CH_2_), 3.0–1.5 (m, 10H, B_10_H_10_); ^13^C-NMR (DMSO-d_6_, 150.95 MHz): *δ* (ppm) = 169.58 (1C, C_amide_), 163.31 (1C, C_imide_), 163.08 (1C, C_imide_), 137.74–120.69 (10C, 10C_arom_), 75.84 (1C, C_carborane_), 56.42 (1C, CH_carborane_), 53.71 (1C, N-*C*H_2pyrrolidine_-CH_2_), 52.93 (1C, CH_2_-pyrrolidine), 38.57 (1C, CH_2_-N(CO)_2_), 36.16 (1C, C*H*_2_-CONH), 31.14 (1C, CH_2_-carborane), 23.16 (1C, CH_2_-*C*H_2pyrrolidine_-CH_2_); ^11^B-NMR {^1^H BB} (DMSO-d_6_, 192.59 MHz): *δ* (ppm) = −4.58 (s, 1B, B5), −11.09 (m, 5B, B4, 6, 9, 10, 12), −13.55 (s, 2B, B8, 11), −14.94 (s, 2B, B2, 3); UV (99.8% EtOH): λ_max_ = 254, 338, 375 nm, λ_min_ = 293, 358 nm, λ_sh_ = 229 nm; FT-IR: *ν*_max_ (cm^−1^) = 2921 (C-H_aliphat_), 2597 (B-H), 1696 (C=O), 1654 (C=O), 728 (B-B); RP-HPLC (gradient B): *t*_R_ = 21.27 min; APCI-MS: *m*/*z*: 508 [M + H]^+^, calcd for C_23_H_33_B_10_N_3_O_3_ = 507.

### 3.2. Biological Investigation

#### 3.2.1. Cytotoxicity Assay

The cytotoxic properties of synthesized compounds were evaluated using human cancer cell line HepG2 established from hepatocellular carcinoma. Cell lines was purchased from Leibniz Institute DSMZ-German Collection of Microorganisms and Cell Cultures (DSMZ, Braunschweig, Germany) and from ECACC (Salisbury, UK).

HepG2 cells were propagated in Dulbecco′s Modified Eagle Medium (DMEM, Braunschweig, Germany; Life Technologies, Warsaw, Poland) supplemented with 10% heat-inactivated fetal bovine serum (FBS; Life Technologies, Warsaw, Poland) and 100 units/mL penicillin G with 100 mg/mL streptomycin (Life Technologies, Warsaw, Poland). The cells were incubated at 37 °C in a humidified atmosphere containing 5% CO_2_.

Upon reaching 80–90% confluency, cells were harvested with 0.25% trypsin in 1 mM EDTA (Life Technologies, Warsaw, Poland) and transferred into 96-well microplates at 5000 cells/well (HepG2 cells). After overnight incubation of cells at 37 °C in a humidified atmosphere containing 5% CO_2_, the culture medium was removed and replaced with a freshly prepared solution of compounds in culture medium or medium itself as the control group.

Stock solution of each compound was prepared in DMSO at 50 mM. Stock solutions were diluted with the growth medium supplemented with 5% FBS to ensure drug dissolution for obtaining concentrations ranging from 0.1 to 200 μM. The cytotoxicity was evaluated by the MTT assay. The final content of DMSO in solutions did not exceed 0.2%, and an additional control group with 0.2% DMSO was included to rule out the effect of solvent. After treatment, cells were incubated at 37 °C in 5% CO_2_ for additional 24 h. Upon completion of the incubation, the medium was aspirated and replaced with 3-(4,5-dimethylthiazol-2-yl)-2,5-diphenyltetrazolium bromide (MTT) dye solution (50 µL, 0.5 mg/mL). After 3 h incubation, the resulting MTT formazan crystals were dissolved in DMSO (100 µL). To ensure the complete dissolution of formazan, the plates were shaken on an orbital microplate shaker at 1000 rpm for 15 min (Thermoshaker NeoLab 7–0055, Bionovo, Legnica, Poland). The optical density of each well was then measured on a microplate spectrophotometer (Bio-Rad 680; Bio-Rad, Warsaw, Poland) at a wavelength of 570 nm. Each experiment consisted of 5 replications of each concentration, and each experiment was repeated three times independently. The results were calculated as percentage of control group viability. The IC_50_ values were determined using a nonlinear regression from the plot of % viability against log dose of compounds by using GraphPad Prism 6.0 software (GraphPad, San Diego, CA, USA).

#### 3.2.2. Cell Cycle Analysis by Flow Cytometry

HepG2 (5 × 10^5^) cells were seeded onto 6-well cell culture plates and incubated for 24 h with the analyzed compounds at a concentration corresponding to whole IC_50_ values. DNA content was determined by flow cytometry with PI (Sigma-Aldrich (Steinheim, Germany)) staining. After incubation, the cells were trypsinized and washed twice with PBS (1 mL). In the next step, the cells were fixed with an ice-cold 80% ethanol. After 1 h incubation in 4 °C, the cells were stored at −20 °C for further analysis. After double wash with PBS, the fixed cells were stained with PI (50 μg/mL) with the addition of RNAse A (100 μg/mL) for 30 min at 37 °C in the dark. The PI fluorescence was measured by FACSCalibur (Becton Dickinson, Franklin Lake, NJ, USA), and data were analyzed by FlowJo software.

#### 3.2.3. Oxidative Stress Measurements in HepG2 Cells by Flow Cytometry

HepG2 cells (3.5 × 10^5^) were seeded onto 6-well plates, cultured with EMEM media at 37 °C and 5% CO_2_ saturation, and incubated until 60–70% confluence. Subsequently, the cells were treated for 24 h with the tested compounds at a concentration that corresponds to a half of IC_50_ value. Next, the cells were detached with trypsin (Sigma-Aldrich (Steinheim, Germany)) and washed twice with DPBS (1 mL) (Thermo Fisher Scientific (Waltham, MA, USA)), and the level of intracellular ROS generation was analyzed by dual staining with H_2_DCFDA/PI according to the manufacturer′s protocol (Thermo Fisher Scientific (Waltham, MA, USA)), in which fluorescence was triggered in the presence of ROS.

#### 3.2.4. Analysis of 8-Oxo-dG in HepG2 Cells by HPLC-UV-ED

HepG2 cells (1 × 10^6^) were seeded onto flasks (25 cm^2^) and cultured in the supplemented growth medium for 16 h. Subsequently, the cells were treated with compounds **31** (53 μM), **33** (5 μM), **34** (8 μM), **35** (9 μM), and **36** (6 μM) for 24 h. After incubation, the cells were trypsinized, rinsed twice with PBS, and pelleted by centrifugation (200× *g*, 3 min). Total DNA was isolated from the treated cells and untreated control cells by using Genomic mini DNA isolation kit (A & A Biotechnology, Gdynia, Poland) according to the manufacturer′s protocol. Quality of the total DNA was assessed spectrophotometrically.

DNA (1 μg) was dissolved in sodium acetate buffer (10 μL, 40 mM, pH 5.3) containing ZnCl_2_ (0.1 mM) and digested witH-Nuclease *P*1 (3 μg). Samples were incubated for 3 h at 37 °C. Subsequently, Tris–HCl (2 μL, 1M, pH 8.0) and alkaline phosphatase (1 U) were added and incubated 2 h at 37 °C. All DNA hydrolysates were ultrafiltered using cut off 10,000 Da filter units.

8-Oxo-dG, and dG in hydrolysates were determined using HPLC-UV followed by electrochemical detection and ED (Coulochem III Electrochemical Detector, ESA, Inc., Chelmsford, MA, USA). DNA hydrolysates were chromatographed isocratically by using ammonium acetate (50 mM pH 5.3)/methanol (93:7, *v*/*v*). Detection of dG was performed at 254 nm. 8-Oxo-dG was performed by the electrochemical detector: guard cell + 400 mV, detector 1: +130 mV (as a screening electrode), detector 2: +350 mV (as a measuring electrode set to sensitivity of 500 nA/V). All results are expressed as mean ± SD.

To determine 8-oxo-dG contents guanosine amount was necessary. The total number of 8-oxo-dG in genome was calculated using special formula [80].

#### 3.2.5. Apoptosis/Necrosis Assay by Flow Cytometry

Apoptosis/necrosis assay was performed by double staining of cells with YO-PRO-1 (Thermo Fisher Scientific (Waltham, MA, USA)) and propidium iodide (PI, Sigma-Aldrich (Steinheim, Germany)) fluorescent dyes. Briefly, HepG2 cells (0.5 × 10^5^) were seeded onto 6-well plates. On the next day, the cells were treated for 24 h with the analyzed compounds at a concentration corresponding to whole IC_50_ values. Subsequently, the cells were detached with trypsin (Thermo Fisher Scientific (Waltham, MA, USA)), washed twice with DPBS (1 mL) (Thermo Fisher Scientific (Waltham, MA, USA)), and stained with YO-PRO-1 and PI according to the manufacturer′s protocol for 30 min at 37 °C in the dark. The cells were analyzed immediately after staining with 488 nm excitation by Accuri C6 flow cytometer (Becton Dickinson, Franklin Lake, NJ, USA).

#### 3.2.6. Apoptosis Detection Using Annexin V Conjugate Staining

HepG2 cells (4 × 10^5^) were seeded onto 6-well plates, cultured with EMEM media at 37 °C and 5% CO_2_, and incubated until 60–70% confluence. Subsequently, the cells were treated for 24 h with the tested compounds **31**, **33**–**36** were added at concentration that corresponds to the whole IC_50_ value. After 24 h incubation, the cells were washed with ice-cold PBS buffer (Gibco, Grand Island, NJ, USA), harvest with 1 × trypsin/EDTA (Corning), and centrifuged (1200 rpm, 3 min). Next, the cells were washed in ice-cold PBS and recentrifuged. Supernatants were discarded, and the cells were suspended in annexin-binding buffer (100 µL, 10 mM HEPES, 140 mM NaCl, and 2.5 mM CaCl_2_, pH 7.4). The cells were stained by adding of the Annexin V Alexa Fluor 647 conjugate (5 µL) (Thermo Fisher (Waltham, MA, USA)). The cells were incubated at RT, for 15 min, in the dark. After incubation, annexin binding buffer (400 µL) was added to the Alexa Fluor 647-stained cells, mixed gently, and kept on ice until analysis. The rate of apoptosis was evaluated immediately after incubation by FACSCalibur flow cytometer with excitation at 635 nm.

#### 3.2.7. Autophagy Assay by Flow Cytometry

Autophagy assay was performed in HepG2 cells by staining with Green Detection Reagent (Autophagy Detection Kit, ab139484, Abcam, Cambridge, UK) fluorescent dye with excitation/emission at 463/534 nm. Briefly, the cells (3.85 × 10^5^) were seeded onto 6-well plates containing the growth medium (EMEM) and incubated until 70–80% confluency. On the next day, the cells were treated for 24 h with the analyzed compounds at a concentration corresponding to whole IC_50_ values. Rapamycin (autophagy inducer) (1 µM) was used to create a strong, single positive green fluorescence signal. Subsequently, the cells were detached with trypsin (Thermo Fisher Scientific (Waltham, MA, USA)), washed twice with DPBS (1 mL) (Thermo Fisher Scientific (Waltham, MA, USA)), and stained with Green Detection Reagent according to the manufacturer′s protocol for 30 min at 37 °C in the dark. The cells were analyzed immediately after staining with 488 nm excitation by Accuri C6 flow cytometer (Becton Dickinson, Franklin Lake, NJ, USA).

#### 3.2.8. Lipid Peroxidation Measurements by Flow Cytometry

HepG2 cells (3.85 × 10^5^) were seeded onto 6-well plates, cultured with EMEM media, at 37 °C and 5% CO_2_ saturation, and incubated until 60–70% confluence. Subsequently, the cells were treated for 24 h with the tested compounds at a concentration that corresponds to whole IC_50_ values. Next, the cells were detached with trypsin (Sigma-Aldrich (Steinheim, Germany)), washed twice with DPBS (1 mL) (Thermo Fisher Scientific (Waltham, MA, USA)) and intracellular oxidation of lipids was analyzed by staining with BODIPY^®^ 581⁄591 C11 reagent according to the manufacturer’s protocol (Thermo Fisher Scientific (Waltham, MA, USA)). Upon oxidation in live cells, the reagent shifts fluorescence emission peak from 590 nm (red) to 510 nm (green). Cumene peroxide (10 µM) was used as a positive control to induce lipid peroxidation. Cells were analyzed immediately after staining, with 488 nm excitation by FACSCalibur flow cytometer (Becton Dickinson, Franklin Lake, NJ, USA) and data were analyzed by FlowJo software. The ratios of the signal from the 590 to 510 channels were used to quantify lipid peroxidation in cells.

#### 3.2.9. Fluorescence Imaging Experiment

HepG2 wells were seeded on a glass-bottom 4-well CELLview cell culture dishes (Greiner Bio-One GmbH, Kremsmünster, Austria) at a density of 10 × 10^4^/well and cultured in supplemented EMEM until 80–90% confluency. Then cells were treated with tested compounds at the final concentration corresponding to the whole IC_50_ value, for 1 h and 24 h. In the next step, lysosomes were labeled with 100 nM of LysoTracker Red DND-99 (Thermo Fisher Scientific (Waltham, MA, USA)) for 15 min, and nuclei were stained with 3 μg/mL of Hoechst 33342 (Thermo Fisher Scientific (Waltham, MA, USA)) for 5 min. After incubation, the cells were gently rinsed in PBS to remove free dyes, placed in the FluoroBrite DMEM (Thermo Fisher Scientific (Waltham, MA, USA)). Live cell imaging was performed using a Leica TCS SP5 II confocal laser scanning microscope equipped with a White Light Laser (470–670 nm), a 405 laser and an environmental cell culture chamber that provided controlled conditions of temperature, CO_2_ saturation and humidity. Fluorescence images were collected using a Plan Apo 63x 1.4 NA oil-immersion objective at Ex/Em 488/500–600 nm for autofluorescence of tested compounds, 561/585–655 nm for lysosomes labeling and 405/430–480 nm for nuclei staining. LAS AF (Leica, Wetzlar, Germany) and Leica LAS X software with a deconvolution module were used for image processing and fluorescence analysis, respectively.

#### 3.2.10. Human Topoisomerase IIα Relaxation Assay

The Human Topoisomerase II-alpha Relaxation Assay Kit was purchased from Inspiralis (Norwich, UK). The Topo II-alpha inhibition assay was performed as described be the manufacturer. Briefly, the reaction mixture (30 μL) containing tested compound (100 µM) dissolved in DMSO (which final concentration of 2% or 4% had no influence on the Topo II-alpha activity), supercoiled pBR322 (0.5 μg) in 1X Assay Buffer and ATP (1 mM) was incubated at 37 °C for 15 min. Next, Topo-II alpha (1 U) was added and the reaction mixture was incubated at 37 °C for an additional 45 min. The reaction was terminated by addition of the STEB buffer (40% (*w*/*v*) sucrose, Tris-HCl (100 mM, pH 8), EDTA (1 mM), bromophenol blue (0.5 mg/mL)). The products were analyzed by electrophoresis using agarose gel (1%) in TEA buffer at 70 V for 2 h, followed by the ethidium bromide staining For the most active compounds **6** and **7** the reaction was repeated in a concentration range of 25–200 µM. The percentage of Topo-II alpha activity inhibition was calculated by densitometric quantification (Quantity One software, Bio-Rad, Warsaw, Poland). The occurrence of a band representing supercoiled DNA on an agarose gel indicated the inhibition of enzyme activity.

#### 3.2.11. Statistical Analysis

Statistical analyzes were performed using GraphPad Prism version 6.0 for Windows, (GraphPad Software, San Diego, CA, USA). Apoptosis and necrosis as well as cell cycle assay was analyzed by two-way analysis of variance (ANOVA), followed by Tukey′s multiple comparison test. ROS induction was analyzed by one-way ANOVA followed by Tukey′s multiple comparison test. Statistical significance for mitochondrial ROS production was determined by one-way ANOVA followed by Dunnett′s multiple comparison test. The results are presented as mean ± SEM from three independent experiments; *p* values less than 0.05 were considered statistically significant. Statistical significance is indicated with asterisks: (*ns*) *p* > 0.05, (*) *p* < 0.05, (**) *p* < 0.01, (***) *p* < 0.001, and (****) *p* < 0.0001.

### 3.3. Physicochemical Investigation with DNA

#### 3.3.1. Materials

Calf-thymus (ct-DNA) was purchased from Sigma (St. Louis, MO, USA) and used without purification. Sodium cacodylate (for the preparation of cacodylate buffer) was purchased from Acros Organics (Geel, Belgium). Water was obtained from a Milli-Q purification system. All experiments were performed with freshly prepared solutions.

#### 3.3.2. Preparation of ct-DNA

The ct-DNA was dissolved in H_2_O, reconstituted overnight at 4 °C to dissolve all the material, and then filtered through a 0.45-μm filter. The molar concentration of ct-DNA was determined from UV-visible spectra by using molar absorption coefficient (*ε*) of 6600 M^−1^ cm^−1^at 260 nm [81]. The purity of ct-DNA was confirmed by UV-visible spectroscopy by measuring the ratio of absorbance at 260 nm to 280 nm and was found to be ≥1.8, indicating that DNA was sufficiently free of proteins.

#### 3.3.3. Melting Temperature (*T*_m_) Measurements

The measurements were performed by adding aliquots of acetone stock solution of the tested compounds to the buffer solution (pH 7.0, 20 mM, cacodylate buffer, acetone content of the final solution = 0.23–0.35%). The *T*_m_ curves were collected at r = 0.3 (r = [compound]/[ct-DNA]) to assure the dominant binding mode. Thermal melting curves were determined by following the absorption change at 260 nm as a function of temperature by using a GBC Cintra10 UV-VIS spectrometer equipped with a GBC Thermocell Peltier Power Supply using a 1 cm path length cell. The absorbance of the samples was monitored at 260 nm from 30 °C to 90 °C with a heating rate of 1 °C/min. *T*_m_ values are the midpoints or the transition curves determined from the maximum of the first derivative. The Δ*T*_m_ values were calculated by subtracting *T*_m_ of the free nucleic acid from *T*_m_ of the sample. Every Δ*T*_m_ value reported in the study was the average of at least three measurements. The error in Δ*T*_m_ was ±0.5 °C.

#### 3.3.4. Circular Dichroism Measurements

The measurements were performed by adding aliquots of DMSO stock solution of the tested compounds to the buffer solution (pH 7.0, 20 mM, cacodylate buffer, DMSO content of the final solutions = 0.38–0.58%). Changes in the CD spectrum of ct-DNA upon the addition of compound were measured at different molar ratios r = [compound]/[ct-DNA]. Circular dichroism spectra were recorded on a JASCO J-815 CD spectrometer with a JASCO CDF-426S Peltier thermostated cell holder (JASCO, Tokyo, Japan) by using a rectangular quartz cuvette of path length 0.5 cm (1 mL) in the 230–400 nm region. The reported CD profiles are an average of three successive scans with 200 nm per minute scan time and an appropriately corrected baseline. The temperature was maintained at 20 °C during the experiment.

#### 3.3.5. Ultraviolet-Visible Spectra Titration

The measurements were performed by adding aliquots of acetone stock solution of the tested compounds to the buffer solution (pH 7.4, 20 mM, 50 mM NaCl, Tris-HCl buffer, acetone content of the final solutions = 0.15–1.15%) to the final concentration 10 µM. The tested compounds were incubated 5 min, with increasing concentrations ranging from 0 to 15 µM of ct-DNA at 37°, then the UV-vis absorption spectra, between 315–455 nm, were recorded using a GBC Cintra10 UV-VIS spectrometer equipped with a GBC Thermocell Peltier Power Supply using a 1 cm path length cell. The binding constant was calculated according to the following equation [82]:(1)$$\frac{A_{0}}{A-A_{0}}=\;\frac{\varepsilon_{G}}{\varepsilon_{H-G}-\varepsilon_{G}}+\;\frac{\varepsilon_{G}}{\varepsilon_{H-G}-\varepsilon_{G}}\;\times \;\frac{1}{K_{b}\left[\mathrm{DNA}\right]}$$
where K*_b_* is the binding constant, *A*_0_ and a are absorbance of the free tested compound and the apparent one, ε_G_ and ε_H−G_ are their coefficient respectively, [DNA] is the concentration of [DNA] in base pair. The slope to intercept ratio of the plot between *A*_0_/*A*−*A*_0_ versus 1/[DNA] yielded the binding constant.

### 3.4. Theoretical Calculations

#### 3.4.1. Model Building and CoMFA Modeling

Sybyl-X 2.0/Certara and HyperChem 6.0 programs were engaged to conduct the molecular modeling simulations. OpenBabel (inter)change file format converter was employed for data conversion. The crystallographic geometry of carborane cluster was retrieved from Crystallographic Data Centre (deposition code: 1010195) and modified accordingly. The initial compound geometry optimization was performed using the optimized potentials for liquid simulations (OPLS) force field and Polak-Ribiere (conjugate gradient) method implemented in HyperChem 6.0 with a 0.1 kcal/A mol energy gradient convergence criterion. The specification of the electrostatic potential values based on the partial atomic charges was carried out with the Gasteiger–Hückel method implemented in Sybyl-X. CoMFA modeling efficiency of electronic and steric potentials in the molecular environment is directly controlled by the specification of the atomic superimposition. Hence, one 6-ordered atom trial alignment on the most active molecule **9** according to the active analogue approach (AAA) was used in FIT method to cover the entire bonding topology of naphtalimide ring (pharmacophore hypothesis). The steric and electrostatic potentials were calculated using sp^3^ carbon probe atom with a charge of +1. CoMFA grid spacing was 2.0 Å for all the dimensions within the defined region, which extended beyond the van der Waals envelopes of all molecules by at least 4.0 Å—for each molecule the energies with a total of 1680 grid points were calculated with 2 Å spacing in a lattice of 14 × 12 × 10, respectively. The potential energies at each lattice point can be plotted as three-dimensional color-coded contour maps indicating the regions where steric hindrance and/or charged substituents enhance or diminish the binding affinity.

#### 3.4.2. Similarity-Based Activity Landscape (SALI)

The numerical sampling of similarity-related structure-activity landscape index (SALI) can be quantitatively expressed according to the following formula:(2)$$SALI_{x,y}=\frac{\left|A_{x}-A_{y}\right|}{1-sim\left(x,y\right)}$$
where $A_{x}$ and $A_{y}$ are the activity profiles for the *x*-th and *y*-th molecule and *sim*(*x*,*y*) is the pair-wise similarity evaluation [83,84]. Tanimoto coefficient was engaged for the fingerprint-based similarity estimation, where the structural pair-wise molecular relatedness is approached using the equation:(3)$$T\left(x,,y\right)=\frac{n_{xy}}{\left(n_{x}+n_{y}-n_{xy}\right)}$$
where$\;n_{xy}$ is the number of bits set into 1 shared in the fingerprint of the molecule *x* and *y*, $n_{x}$ is the number of bits set into 1 in the molecule *x*, $n_{y}$ is the number of bits set into 1 in the molecule *y*, respectively.

#### 3.4.3. Principal Component Analysis (PCA) and Partial Least Squares Method (PLS)

The human-friendly 2D/3D plots of the compound distribution in the experimental-based (FCS) and virtual-derived (VCS) molecular space might be displayed using the principal component analysis (PCA) method. PCA is a linear projection method, that can be applied to model multivariate data with a relatively small number of so-called principal components (scores and loadings) generated to maximize the description of variance within the input data [85]. The PCA model with *f* principal components for a data matrix *X* can be calculated as follows:(4)$$X=\;TP^{T}+E$$
where *X* is a data matrix with *m* objects and *n* variables, *T* is the score matrix with dimensions (*m* × *f*), *P^T^* is a transposed matrix of loadings with dimensions (*f* × *n*) and *E* is a matrix of the residual variance (*m* × *n*) not explained by the first *f* principal components. On the whole, the first few principal components (PCs) frequently describe sufficiently data variance and reveal the groups of objects.

The partial least squares (PLS) approach generates a regression relation between variable Y and an ensemble of descriptors **X** according to the following equation:(5)Y= Xb+e
where *b* is a vector of the regression coefficients and *e* is a vector of the errors. The complexity of PLS models was estimated using the leave–one–out cross–validation procedure (LOO-CV) [86]. The dependent variable for each left–out object is calculated based on the model with one, two, three, etc. factors, respectively. A cross–validated leave–one–out $q_{cv}^{2}$ metric for the approximation of the model performance is calculated as follows:(6)$$q_{cv}^{2}=1-\frac{\sum_{i}^{m}{\left(obs_{i}-pred_{i}\right)}^{2}}{\sum_{i}^{m}{\left(obs_{i}-mean\left(obs_{i}\right)\right)}^{2}}$$
where *obs* is the assayed value; *pred* is the predicted value; *mean* is the mean value of *obs*, and *i* refers to the object index ranging from 1 to *m*. The cross-validated standard error of prediction *s* is estimated using the equation:(7)$$s=\sqrt{\frac{\sum_{i}^{m}{\left(obs_{i}-pred_{i}\right)}^{2}}{m-k-1}}$$
where *m* is the number of objects and *k* is the number of the PLS factors in the model. The quality of external predictions was validated by the standard deviation of error of prediction (SDEP), $q_{test}^{2}$ and the mean absolute error (MAE) metrics defined as:(8)$$SDEP=\sqrt{\frac{\sum_{i}^{n}{\left(pred_{i}-obs_{i}\right)}^{2}}{n}}$$
(9)$$q_{test}^{2}=1-\frac{\sum_{i}^{n}{\left(obs_{i}-pred_{i}\right)}^{2}}{\sum_{i}^{n}{\left(obs_{i}-mean\left(obs_{i}\right)\right)}^{2}}$$
(10)$$MAE=\frac{\sum_{i}^{n}\left|pred_{i}-obs_{i}\right|}{n}$$
where n is the number of molecules in the test set.

## 4. Conclusions

In this paper, we have described convenient protocols for synthesizing 1,8-naphthalimide derivatives modified with an *ortho*- or *meta*-carborane clusters at position 3 of the heteroaromatic skeleton. The X-ray structure of the 1,8-naphthalimide–carborane conjugates **39** and **41** was established. In addition, we determined the cytotoxic activity of the modified conjugates against HepG2 cells and found that modified naphthalimides were significantly more active than modified anhydrides. Conjugates modified with *ortho*-carboranes were rather more active than those bearing *meta*-carboranes. The type of linker chain between the carborane cluster and the heterocyclic system influenced the antiproliferative activity of the naphthalimides, and the activity varied as follows: **8**–**11** (-triazole- linker) > **33**–**36** (-NH-CH- linker) > **39**–**42** (-NH-CO- linker) > **17**–**20** (-O-triazole- linker). Furthermore, our study showed that modified conjugates could effectively induce cell cycle arrest at G0/G1 or G2M phase and activate mainly apoptosis as well as autophagy and ferroptosis which was confirmed using the flow cytometry analysis. Among the studied compounds, conjugate **35** induced ROS production resulting in almost four times higher number of 8-oxo-dG in comparison to mitonafide, and strongly promoted apoptosis which might inhibit the growth of HepG2 cells. However, the presence of the carboranyl cluster at position 3 of these compounds did not promote them as effective Topo II inhibitors.

The DNA binding properties of the synthesized compounds were also investigated by UV–vis, CD, and thermal denaturation experiments. It was found that these compounds were rather weak classical DNA intercalators, which indicated another type of interaction with DNA. The lysosome-targeting behavior of compounds **33**, **34**, and **36** and their imaging capacity w studied by co-localization experiments, which revealed that these conjugates selectively localized in the lysosomes, which additionally enabled the localization of boron/carborane in the cells.

We assessed the similarity-driven property space for the series of carborane-containing conjugates using PCA. The investigation of the spatial space defined by the first three orthogonal components (PC1–PC3) revealed that carborane-based derivatives are clustered into four subgroups. The projection of the pIC_50_ values on the PC1 vs. PC2 plane clearly indicated the diagonal separation of the active and nonactive conjugates. Moreover, the enhancement of the planar descriptor-driven projection with response data resulted in a structure–activity landscape that can be regarded as a subtle picture of (dis)allowed structural adjustment(s) potentially valid for molecular activities. Interestingly, the replacement of the anhydride-like fragment with the imide-based motif influenced the molecular potency, as shown by the comparison of the most active (**11**, **33**, **8**) and nonactive (**6**, **7**, **15**, **16**) compounds, respectively. Finally, we performed a quantitative CoMFA ligand-based study to specify the potentially valid steric and electrostatic features of the pharmacophore pattern. In this case, the bundle of steric bulk was indicated as the privileged feature contributing (un)favorably to the CoMFA model that was validated accordingly.

The quantitative comparison of the field-based descriptors can lead to a useful picture of the drug–receptor recognition pattern; however, it should be treated as a crude approximation of the underlying biological reality.

The present work demonstrated that new organic and inorganic hybrids can be considered as a novel class of compounds with potential antitumor activities. Studies on the synthesis and biological properties of new naphthalimides bearing modified carborane at position 4 of the naphthalic ring are carried out in our laboratory and will be published soon.

## Data Availability

Not applicable.

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
