# Peer review of "Design, Synthesis, and Evaluation of Novel 3-Carboranyl-1,8-Naphthalimide Derivatives as Potential Anticancer Agents"

_ijms, 2021, doi:10.3390/ijms22052772_

Round 1

Reviewer 1 Report

This paper fits the scope of the journal. The authors synthesized 28 novel 3-carboranyl-naphthalimide derivatives testing their in vitro cytotoxicity (HepG2 cell line), cell cycle, ROS production, cell death (apoptosis, autophagy, and ferroptosis induced by compounds after 24 h of treatment), lysosome-targeting behavior, biophysical studies on DNA interaction, and structure-activity relationship studies.

The compound’s characterizations are precise and convincing, highlighting the immense work and the dedication of the authors for this project. The manuscript appears well-written and full of details, that allow the understanding of each step, even to non-experts. It is composed in a clear style and, in general, the procedures employed seem rational.

However, there are some small issues with the manuscript, which lead me to conclude that this work is not suitable for publication in IJMS, MDPI, in its current form and some changes would need to be implemented before the publication.

Major point:

From Figure 9, Mitonafide turns out to be the most potent ROS inducer, very different from the other compounds analyzed. On the contrary, compounds 31, 33-36 reported in Table 2 resulted comparable to Mitonafine reference (with 35, around 4 times more active than the reference).

Despite the compounds of Fig.9 did not exhibit autofluorescence at Ex/Em 488/500-600 nm, I suggest to the authors testing the most active compounds of Fig.9 (as 10, 18, 32, 39) measuring the level of ROS production as 8-oxo-dG content. These results could allow the authors to better compare the results on ROS production.

Line 621 – Figure 11 is really complicated to fully understand. Given the result of the authors, is it not possible to add an enlargement of some sections?

Line 680 – The melting temperature of compounds/ct-DNA interactions was evaluated at 260 nm. Reading the experimental procedure, it is not clear if the authors have considered that their compounds have a maximum of Abs at 260 nm (it was not possible to calculate the extinction coefficient from UV spectra reported in ESI because the concentrations are missing, please add them). Thus, is it possible that the ε values of compounds are higher than ct-DNA (6600 cm-1M-1, line 1669) that no significant UV changes are seen?

Line 727 – All UV–vis absorption titration spectra, reported in ESI, reported changing within the Abs range of 0.01-0.03. Moreover, it is not clear whether the spectra end on the zero of Abs, from the images shown. If not, how the authors are sure that the changes were due to interactions and not to precipitation of ct-DNA or ct-DNA/compounds complexes?

Minor points:

Line 117 – Please correct the article with “a SAR-mediated”.

Line 132 – The verb “was” does not agree with the two subjects: Ascorbate and Copper. Please change with the plural form “were”.

Line 173 – I suppose that “1-(3-azidopropyl)-1,7-dicarba-closo-dodecaborane” is compound 5, not “for 5”. Please check the caption of Figure 2.

Line 287 – To maintain the manuscript format, I suggest adding the name of compounds 37 and 38 in the caption of Figure 5, as the authors did for the captions of the previous figures.

Line 366 – Considering the standard deviation, IC50 values of compounds 8 and 33 are comparable. I suggest the authors modifying the sentence “the highest activity among the mitonafide analogs modified with a carborane cluster (10, 17, 19, 33, 35, 39, 41)”, removing compound 33.

Line 383-384 – From Table 1, it is clear that the authors are referring to compound 33, nevertheless, I suggest adding it to the sentence, to maintain the manuscript format.

Line 403 – Remove “the” after HepG2.

Line 443 – Caption of Fig. S156. Please replace “what” with “which”.

Line 511 - I suggest to the authors moving Table S1 after Fig. S158 (and not at the end of ESI), to make data analysis more fluid. Moreover, except for compounds 7, 17, and Pinafide, the percentage summary is not 100.00% in Table S1. Please check the data or add standard deviation values.

Line 530 – I suppose that the comma after “Through this” is not necessary. Please check.

Line 580 - Please check the caption of Fig. S161. I suppose that the right compound is “cumene hydroperoxide”.

Line 675 - It is not clear why the authors decided to evaluate the interaction with DNA for compounds 33, 34, and 36 if they previously verified their specific targeting to lysosomes.

Line 906. I suppose that the right sentence is “The performance of this CoMFA model”. Please, check.

1601 – Please remove “of” before “DPBS”.

Author Response

Reviewer 1

This paper fits the scope of the journal. The authors synthesized 28 novel 3-carboranyl-naphthalimide derivatives testing their in vitro cytotoxicity (HepG2 cell line), cell cycle, ROS production, cell death (apoptosis, autophagy, and ferroptosis induced by compounds after 24 h of treatment), lysosome-targeting behavior, biophysical studies on DNA interaction, and structure-activity relationship studies.

The compound’s characterizations are precise and convincing, highlighting the immense work and the dedication of the authors for this project. The manuscript appears well-written and full of details, that allow the understanding of each step, even to non-experts. It is composed in a clear style and, in general, the procedures employed seem rational.

However, there are some small issues with the manuscript, which lead me to conclude that this work is not suitable for publication in IJMS, MDPI, in its current form and some changes would need to be implemented before the publication.

Major point:

From Figure 9, Mitonafide turns out to be the most potent ROS inducer, very different from the other compounds analyzed. On the contrary, compounds 31, 33-36 reported in Table 2 resulted comparable to Mitonafine reference (with 35, around 4 times more active than the reference).

We appreciate Reviewer’s remark. Mitonafide was the most potent ROS inducer among the compound shown in Figure 9. Mitonafide was not the most potent ROS inducer of all naphthalimide-carborane conjugates. Therefore, we have made changes in our manuscript (page 12):

was: “The most potent ROS inducer was mitonafide, whereas naphthalimide-carborane conjugates 10, 18, 32, and 39 were less effective.”

is: “The most potent ROS inducer, among conjugates 6-11, 15-20, 32, 39-42, was mitonafide. Conjugates 10, 18, 32, and 39 were less effective.”

Despite the compounds of Fig.9 did not exhibit autofluorescence at Ex/Em 488/500-600 nm, I suggest to the authors testing the most active compounds of Fig.9 (as 10, 18, 32, 39) measuring the level of ROS production as 8-oxo-dG content. These results could allow the authors to better compare the results on ROS production.

We agree with the Reviewer that ROS production and related oxidative stress is a very important issue in biological systems and it is very important to understand precisely the influence of analyzed compounds on ROS level in the cells. Flow cytometric analysis that we performed showed that the analyzed compounds (6-11, 15-20, 32, 39-42) did not increase ROS level significantly. The compounds chosen by the Reviewer for further analysis of 8-oxo-dG contents after the treatment (10, 18, 32, 39) although they are the most active within the analyzed group, are similar ROS inducers to others, increasing mean fluorescence intensity compared to control as follows:

Compound

mean fluorescence intensity [RFU]

Control

381

10

592.5

18

576

32

610.5

39

576

6

421.25

7

452

8

469.5

9

542.5

11

483.5

15

477

16

500

17

537.5

19

472.5

20

528.5

40

497

41

523.5

42

507

These results prompted us to the detailed analysis of ROS level as the 8-oxo-dG contents only in the compounds presenting the most promising biological properties (31, 33, 34, 35, and 36).

Line 621 – Figure 11 is really complicated to fully understand. Given the result of the authors, is it not possible to add an enlargement of some sections?

Figure 11 has been enlarged. Individual sections in each photo have been enlarged and are located in an additional panel (Zoom).

Line 680 – The melting temperature of compounds/ct-DNA interactions was evaluated at 260 nm. Reading the experimental procedure, it is not clear if the authors have considered that their compounds have a maximum of Abs at 260 nm (it was not possible to calculate the extinction coefficient from UV spectra reported in ESI because the concentrations are missing, please add them). Thus, is it possible that the ε values of compounds are higher than ct-DNA (6600 cm-1M-1, line 1669) that no significant UV changes are seen?

According to the law of absorption, the dependence of absorbance on concentration should be linear. However, in practice, you should take into account exceptions to such a dependence. Deviations from the laws of absorption are caused by: 1) the basic limitations of laws, 2) chemical factors, 3) apparatus factors. The absorption laws are fulfilled for dilute solutions (c < 10-2 M). In such solutions, the extinction coefficient does not depend on the refractive index of the light.

We did not measure the extinction coefficient of our compounds. It can be rationally used that the molar extinction coefficient does not depend on the temperature.

The absorbance of double-stranded nucleic acids is different from the absorbance of the sum of their single-stranded components. The measurement of the melting temperature of nucleic acids is based on this, therefore it can be assumed with a high degree of probability that the observed softening curves correspond to the degree of association of complementary nucleic acids under the influence of the tested compounds, even if the absolute absorbance values may contain the absorbance component of the tested compounds.

Line 727 – All UV–vis absorption titration spectra, reported in ESI, reported changing within the Abs range of 0.01-0.03. Moreover, it is not clear whether the spectra end on the zero of Abs, from the images shown. If not, how the authors are sure that the changes were due to interactions and not to precipitation of ct-DNA or ct-DNA/compounds complexes?

We didn't observe any precipitation in the cuvette during titration. Potential precipitation of ct-DNA and compound-complex would show in later phases of the experiment when higher concentrations of ct-DNA are used and where more complex formed is present in solution. We observe the bigger effect interaction of the ligand with ct-DNA in earlier stages (low concentrations of ct-DNA), which is characteristic for interactions between ct-DNA and compound. To exclude the effect of putative precipitation of the compound during analysis, we routinely incubate the solution of the compound in a buffer for several minutes before the experiment and check if there are any changes in the absorbance spectrum.

Minor points:

Line 117 – Please correct the article with “SAR-mediated”.

Corrected

Line 132 – The verb “was” does not agree with the two subjects: Ascorbate and Copper. Please change with the plural form “were”.

Corrected

Line 173 – I suppose that “1-(3-azidopropyl)-1,7-dicarba-closo-dodecaborane” is compound 5, not “for 5”. Please check the caption of Figure 2.

We appreciate Reviewer’s remark. 1-(3-Azidopropyl)-1,7-dicarba-closo-dodecaborane is compound 5. It was corrected. Now is stated: “1-(3-azidopropyl)-1,7-dicarba-closo-dodecaborane (5), CuSO4×5H2O, sodium ascorbate, and THF/H2O, 2 h, 35 °C (for 7)”.

Line 287 – To maintain the manuscript format, I suggest adding the name of compounds 37 and 38 in the caption of Figure 5, as the authors did for the captions of the previous figures.

The name of compound 37 and 38 were added in the caption of Figure 5.

Now is stated: “Figure 5. Modification of naphthalimide derivatives with ortho-/meta-carborane via the formation of amide bond: (i) 3-(1,2-dicarba-closo-dodecaboran-1-yl)propionic acid (37) or 3-(1,7-dicarba-closo-dodecaboran-1-yl)propionic acid (38), PyBOP, CH2Cl2, TEA, 4–6 h, RT. The yield of the compound is given in parentheses.”

Line 366 – Considering the standard deviation, IC50 values of compounds 8 and 33 are comparable. I suggest the authors modifying the sentence “the highest activity among the mitonafide analogs modified with a carborane cluster (10, 17, 19, 33, 35, 39, 41)”, removing compound 33.

We appreciate Reviewer’s remark. Compound 33 was removed, and the sentence was modifying. Now, it is read:

“The mitonafide analog modified with ortho-carborane 8 (IC50 = 4.33 µM) showed a moderately lower cytotoxic activity compared to compound 9, but the highest activity among the mitonafide analogs modified with a carborane cluster (10, 17, 19, 35, 39, 41).”

Line 383-384 – From Table 1, it is clear that the authors are referring to compound 33, nevertheless, I suggest adding it to the sentence, to maintain the manuscript format.

Compound 33 was added to the sentence. Now, it is read:

“Mitonafide analog bearing ortho-carborane 33 was the most active among the synthesized compounds with an IC50 value of 4.77 µM.”

Line 403 – Remove “the” after HepG2.

Corrected

Line 443 – Caption of Fig. S156. Please replace “what” with “which”.

Corrected

Line 511 - I suggest to the authors moving Table S1 after Fig. S158 (and not at the end of ESI), to make data analysis more fluid. Moreover, except for compounds 7, 17, and Pinafide, the percentage summary is not 100.00% in Table S1. Please check the data or add standard deviation values.

Table S1 was moved after Figure S162. As recommended by the second Reviewer, four HRMS spectra were added to ESI. The numbering of the Figures in ESI were adjusted adequately. The total number of figures is 203. Figure S158 was converted to Figure S162.

The presented percentage distribution of live, early, and late apoptotic and necrotic cells in Table S1 were calculated as a mean value of each replicate and rounded to two decimal places. That is why not every sum for each compound gives 100%. To be more precise we added standard deviation values.

Line 530 – I suppose that the comma after “Through this” is not necessary. Please check.

We agree with Reviewer. The comma after “Through this” was removed.

Line 580 - Please check the caption of Fig. S161. I suppose that the right compound is “cumene hydroperoxide”.

We agree with Reviewer. The right compound is cumene peroxide. It was corrected.

Line 675 - It is not clear why the authors decided to evaluate the interaction with DNA for compounds 33, 34, and 36 if they previously verified their specific targeting to lysosomes.

1,8-Naphthalimides are mainly tested as DNA intercalators and anticancer agent. They exert antitumor activity through different pathways described in 1. Introduction part of our manuscript.

Lysosomes are highly dynamic intracellular organelles that involve with the biosynthetic, endocytic, and autophagic pathways (A. Ballabio, J. S. Bonifacino, Lyzosomes as dynamic regulators of the cell and organismal homeostasis, Nature Reviews Molecular Cell Biology, 2020, 21, 101-118). They control the recycling of the majority of cellular macromolecules and organelles through more than 50 acid hydrolases. Cathepsin proteases are the most well studied lysosomal hydrolases and overexpressed in the tumor environment. Lysosomotropic detergents are highly lipophilic weak basis compounds. They are well-known lyzosomal membrane permeabilization inducers (e.g. siramesine). Siramesine accumulates in lysosomes and destabilizes them, inducing lysosomal leakage and cathepsin-dependent death of cancer cells even in the presence of the prosurvival Bcl-2 family of proteins ( Z. Chen et al. A new class of naphthalimide-based anitumor agent that inhibit topoisomerase II and induce lysosomal membrane permeabilization and apoptosis. J. Med. Chem. 2010, 53, 2589-2600, and the papers cited therein).

1,8-Naphthalimide derivatives containing polyamines and hydrocarbon chains were tested as DNA-intercalators, and lysosomotropic detergents (J. Med. Chem. 2010, 53, 2589-2600). It was demonstrated that tested compound were rather weak DNA binders but inhibited topo II modestly. Additionally, they could induce lysosomal membrane permeabilization which was quite consistent with their antiproliferative activity. This proves that naphthalimides can be tested as multitarget drugs.

Line 906. I suppose that the right sentence is “The performance of this CoMFA model”. Please, check.

It was corrected. The right sentence is “The performance of this CoMFA model”.

1601 – Please remove “of” before “DPBS”.

Corrected

Reviewer 2 Report

The submitted paper describes synthesis and biological properties of 1,8-naphthalimide derivatives modified on carbon C3. The content can be divide into three main parts, chemistry, biological investigation and computational calculations. Chemistry part cover introduction of several functional groups (ethynyl, amino, hydroxyl) which in further steps were joined with reactive molecules containing ortho or meta carborane fragment. The last step of applied synthetic strategy involved conversion of anhydrides into appropriate imides. The applied methods of synthesis are well known and involved click reaction or typical condensation. The novelty of this fragment of work is introduction of dodecarboran-1-yl fragment to the 1,8-napthalimide skeleton under optimised conditions.

The second part, biological experiments were very wide planed and fulfil expectation for these type of reports. Particular assessments were performed with usage of model cancer cells line, namely HepG2, according to protocols published by manufacturers. The results show rather moderate activity of synthesized compounds. Among 22 compounds used in cytotoxicity test only four exhibited IC50 at concentration below 5 mM. Explanation of mechanism of action for the synthesized derivatives in cells is a main part of biological tests. Based on obtained results, at present is difficult to indicate unambiguous fashion of action for 1,8-naphthalimide containing carborane cluster. The presented results delivery valuable information about the interaction of derivatives containing boron cluster with nucleic acids and cell organelles. Very nicely was confirmed ability of derivatives to penetrate lysosomes.

The third part consider computational calculation were used for consideration of drug-receptor recognition patterns but at present stage they should be consider rather as a row but useful information.

In sum, I recommend this paper for publication after minor improvements listed below. The presented results are not breath-taking, but deliver new information about synthesis of carborane derivatives and showed how complicated can be interaction between bio-targets and chemical molecules. The results will be interested for other people working with boron derivatives as a potent bioactive compounds.

Suggested changes

  1. Is should be clearly indicated in title, that all obtained compounds are derivatives of 1,8-naphthalimide. Three naphthalimides are recognised by chemists: 1,2; 1,3 and 1,8.
  2. In the first figure I suggest to place structure of dodecaborane with indicated all atoms of a cluster and its equal form used by authors in manuscript. In the rest of text, the applied simplified structure can be used.
  3. In general, for experimental part is declared what spectra are recorded at 700 MHz, 175.95 and 244.50 MHz for proton carbon and boron, respectively, whereas all spectra are recorded at lowest frequency.
  4. I suggest to make a few HRMS-MS spectra for controlling molecular mass. The applied technique APCI-MS gives only approximate results.

Author Response

Reviewer 2

The submitted paper describes synthesis and biological properties of 1,8-naphthalimide derivatives modified on carbon C3. The content can be divide into three main parts, chemistry, biological investigation and computational calculations. Chemistry part cover introduction of several functional groups (ethynyl, amino, hydroxyl) which in further steps were joined with reactive molecules containing ortho or meta carborane fragment. The last step of applied synthetic strategy involved conversion of anhydrides into appropriate imides. The applied methods of synthesis are well known and involved click reaction or typical condensation. The novelty of this fragment of work is introduction of dodecarboran-1-yl fragment to the 1,8-napthalimide skeleton under optimised conditions.

The second part, biological experiments were very wide planed and fulfil expectation for these type of reports. Particular assessments were performed with usage of model cancer cells line, namely HepG2, according to protocols published by manufacturers. The results show rather moderate activity of synthesized compounds. Among 22 compounds used in cytotoxicity test only four exhibited IC50 at concentration below 5 mM. Explanation of mechanism of action for the synthesized derivatives in cells is a main part of biological tests. Based on obtained results, at present is difficult to indicate unambiguous fashion of action for 1,8-naphthalimide containing carborane cluster. The presented results delivery valuable information about the interaction of derivatives containing boron cluster with nucleic acids and cell organelles. Very nicely was confirmed ability of derivatives to penetrate lysosomes.

The third part consider computational calculation were used for consideration of drug-receptor recognition patterns but at present stage they should be consider rather as a row but useful information.

In sum, I recommend this paper for publication after minor improvements listed below. The presented results are not breath-taking, but deliver new information about synthesis of carborane derivatives and showed how complicated can be interaction between bio-targets and chemical molecules. The results will be interested for other people working with boron derivatives as a potent bioactive compounds.

We appreciate Reviewer’s positive opinion on our contribution and his recommendation to publish this work in International Journal of Molecular Sciences after minor revision.

Suggested changes

  1. Is should be clearly indicated in title, that all obtained compounds are derivatives of 1,8-naphthalimide. Three naphthalimides are recognised by chemists: 1,2; 1,3 and 1,8.
  2. In the first figure I suggest to place structure of dodecaborane with indicated all atoms of a cluster and its equal form used by authors in manuscript. In the rest of text, the applied simplified structure can be used.
  3. In general, for experimental part is declared what spectra are recorded at 700 MHz, 175.95 and 244.50 MHz for proton carbon and boron, respectively, whereas all spectra are recorded at lowest frequency.
  4. I suggest to make a few HRMS-MS spectra for controlling molecular mass. The applied technique APCI-MS gives only approximate results.

Ad. 1 We appreciate Reviewer’s remark. The title was changed. Now is stated:

“Design, synthesis, and evaluation of novel 3-carboranyl-1,8-naphthalimide derivatives as potential anticancer agents.”

Ad 2. The structure of dodecaborane with indicated all atoms of a cluster was introduced in Figure 1. Now, the Figure 1 caption is stated:

“General structure of icosahedral dicarba-closo-dodecaborane (closo-carborane, C2B10H12), and example structures of naphthalimides with boron cluster [20].”

Ad 3. We entirely agree with the Reviewer’s opinion. 1H NMR, 13C NMR, and 11B NMR spectra were recorded on a Bruker Avance III 600 MHz spectrometer equipped with a direct ATM probe. The spectra for 1H, 13C, and 11B nuclei were recorded at 600.26 MHz, 150.94 MHz, and 192.59 MHz, respectively. This change was introduced in the revised manuscript (3. Materials and Methods, 3.1. Chemistry):

1H NMR, 13C NMR, and 11B NMR spectra were recorded on a Bruker Avance III 600 MHz spectrometer equipped with a direct ATM probe. The spectra for 1H, 13C, and 11B nuclei were recorded at 600.26 MHz, 150.94 MHz, and 192.59 MHz, respectively.” (page 23).

Ad. 4. HRMS spectra for controlling molecular mass, for compounds 8 (modified with carborane cluster via click reaction), 20 (modified with carborane cluster via click reaction), 35 (modified with carborane cluster via reductive amination), 40 (modified with carborane cluster via amidation reaction) were performed. HRMS spectra were added to ESI:

-compound 8Figure S22. HRMS spectrum of 8 (page 28)

-compound 20 - Figure S86. HRMS spectrum of 20 (page 92)

-compound 35Figure S122. HRMS spectrum of 35 (page 128)

-compound 40Figure S144. HRMS spectrum of 40 (page 150)

The numbering of the remaining Figures in ESI has been adjusted (Figure S1 - S203), and it was also corrected in the manuscript (pages 4, 7, 10, 12-18).

Additionally, HRMS methodology and HRMS spectra describing compounds 8, 20, 35, 40 were added to 3. Materials and Methods, 3.1. Chemistry part:

“High-resolution mass spectra (HRMS) were obtained on a Agilent 6546 LC/Q-TOF with ESI ion source

spectrometer (Agilent Technologies, Inc., Santa Clara, USA)” (page 23).

“HRMS (ESI+) 520.3867 [M+H]+, calcd for C23H33B10N5O2 = 520.3711 [M+H]+ (compound 8, page 26)

“HRMS (ESI+) 576.4156 [M+H]+, calcd for C26H37B10N5O3 = 576.3973 [M+H]+ (compound 20, page 30)

“HRMS (ESI+) 454.3642 [M+H]+, calcd for C20H31B10N3O2 = 454.3493 [M+H]+ (compound 35, page 31)

“HRMS (ESI+) 508.3761 [M+H]+, calcd for C23H33B10N3O3 = 508.3598 [M+H]+ (compound 40, page 33)

Reviewer 3 Report

The manuscript is devoted to synthesis of the new 3 3-carboranyl-naphthalimide derivatives as anticancer agents. The topic is new and promising for readers. The research design is really very strong. The results are excellently made and described. The authors proposed some unique approaches to describe the data and to make conclusions. Summarizing, the article is made at the highest scientific level. It is a seldom situation that I recommend to publish it as it is. 

Author Response

We appreciate very much the Reviewer’s positive opinion on our contribution and his recommendation to publish this work in the International Journal of Molecular Sciences without changes.

Reviewer 4 Report

The manuscript "Design, synthesis, and evaluation of novel 3-carboranyl-naphthalimide derivatives as potential anticancer agents" by Olejniczak and co-workers is described the synthesis of of novel 3-carboranyl-naphthalimide derivatives, mitonafide pinafide analogs, and determination the cytotoxic activity of the modified conjugates against HepG2 cells. The authors showed that the type of linker chain between the carborane cluster and the heterocyclic system influenced the antiproliferative activity of the naphthalimides.

It's a beautiful, well-written work. I have some comments on the paper.

Please add the compounds yield on the figures 2-5 for all synthesized compounds.

Please check the manuscript for potential grammar errors and typos.

Author Response

Reviewer 4

The manuscript "Design, synthesis, and evaluation of novel 3-carboranyl-naphthalimide derivatives as potential anticancer agents" by Olejniczak and co-workers is described the synthesis of of novel 3-carboranyl-naphthalimide derivatives, mitonafide pinafide analogs, and determination the cytotoxic activity of the modified conjugates against HepG2 cells. The authors showed that the type of linker chain between the carborane cluster and the heterocyclic system influenced the antiproliferative activity of the naphthalimides.

It's a beautiful, well-written work. I have some comments on the paper.

We appreciate very much the Reviewer’s positive opinion on our contribution and his recommendation to publish this work in the International Journal of Molecular Sciences without changes.

Please add the compounds yield on the figures 2-5 for all synthesized compounds.

The compounds yield on the Figures 2-5 for all synthesized compounds were added. Additionally, for Figures 2-5 caption, the statement was added: "The yield of the compound is given in parentheses.”

Please check the manuscript for potential grammar errors and typos.

The text of the manuscript was edited for proper English language at Translmed Publishing Group (TPG) (Dallas/Ft. Worth area, 713 Sleepy Hollow Dr., Cedar Hill, TX 75104 U.S.A.) (certificate for request).  
